# Mobilization of LINE-1 retrotransposons is restricted by *Tex19.1* in mouse embryonic stem cells

Marie MacLennan[1†], Marta García-Cañadas[2†], Judith Reichmann[1‡], Elena Khazina[3], Gabriele Wagner[3], Christopher J Playfoot[1], Carmen Salvador-Palomeque[2§], Abigail R Mann[1], Paula Peressini[2], Laura Sanchez[2], Karen Dobie[1], David Read[1¶], Chao-Chun Hung[1], Ragnhild Eskeland[4,5], Richard R Meehan[1], Oliver Weichenrieder[3*], Jose Luis García-Pérez[1,2*], Ian R Adams[1*]

[1]MRC Human Genetics Unit, MRC Institute of Genetics and Molecular Medicine, University of Edinburgh, Edinburgh, United Kingdom; [2]Centro de Genómica e Investigación Oncológica (GENYO), Pfizer-Universidad de Granada-Junta de Andalucía, PTS Granada, Granada, Spain; [3]Department of Biochemistry, Max Planck Institute for Developmental Biology, Tübingen, Germany; [4]Department of Biosciences, University of Oslo, Oslo, Norway; [5]Norwegian Center for Stem Cell Research, Department of Immunology, Oslo University Hospital, Oslo, Norway

*For correspondence: oliver. weichenrieder@tuebingen.mpg. de (OW); Jose.Garcia-Perez@ igmm.ed.ac.uk (JLG-P); Ian. Adams@igmm.ed.ac.uk (IRA)

[†]These authors contributed equally to this work

Present address: [‡]EMBL Heidelberg, Heidelberg, Germany; [§]Mater Research Institute, University of Queensland, Woolloongabba, Australia

[¶]Deceased

**Abstract** Mobilization of retrotransposons to new genomic locations is a significant driver of mammalian genome evolution, but these mutagenic events can also cause genetic disorders. In humans, retrotransposon mobilization is mediated primarily by proteins encoded by LINE-1 (L1) retrotransposons, which mobilize in pluripotent cells early in development. Here we show that TEX19.1, which is induced by developmentally programmed DNA hypomethylation, can directly interact with the L1-encoded protein L1-ORF1p, stimulate its polyubiquitylation and degradation, and restrict L1 mobilization. We also show that TEX19.1 likely acts, at least in part, through promoting the activity of the E3 ubiquitin ligase UBR2 towards L1-ORF1p. Moreover, loss of *Tex19.1* increases L1-ORF1p levels and L1 mobilization in pluripotent mouse embryonic stem cells, implying that *Tex19.1* prevents *de novo* retrotransposition in the pluripotent phase of the germline cycle. These data show that post-translational regulation of L1 retrotransposons plays a key role in maintaining trans-generational genome stability in mammals.

## Introduction

Retrotransposons are mobile genetic elements that comprise around 40% of mammalian genomes (*Beck et al., 2011*; *Hancks and Kazazian, 2016*; *Richardson et al., 2014a*). Retrotransposons are a source of genetic variation that shape genome evolution and mammalian development, but their mobilization can also cause mutations associated with a variety of genetic diseases and cancers (*Beck et al., 2011*; *Hancks and Kazazian, 2016*; *Richardson et al., 2014a*; *Garcia-Perez et al., 2016*). New retrotransposition events are estimated to occur in around 1 in every 20 human births, and represent around 1% of genetic disease-causing mutations in humans (*Kazazian, 1999*; *Hancks and Kazazian, 2016*). Retrotransposons can be classified into two major types depending on their genomic structure and presence of LTR (long terminal repeat) sequences: LINEs (long interspersed elements) and SINEs (short interspersed elements) lack LTR sequences and end in a polyA sequence, while LTR retrotransposons are similar in structure to retroviruses (*Beck et al., 2011*). In humans, all new retrotransposition events are catalysed by LINE-1 (L1) elements. Active L1s encode

**eLife digest** Around half of the DNA in a human cell is made of stretches of genetic code that were copied from one part of the human genome and pasted into new locations. Such moveable pieces of genetic code are known as retrotransposons, most which are no longer active. However, one type can still move around: the 'long interspersed element class 1', called LINE-1 for short.

There are several hundred thousand copies of LINE-1 in the human genome, and each copy encodes two proteins that work together to insert new copies of LINE-1 into the genome. Each time LINE-1 is pasted into a new location, there is a risk that it will disrupt a gene, creating a mutation. If this happens in the cells that make sperm or eggs – known as germline cells – the mutation can be passed on to the next generation. Human cells have some defence against LINE-1. They commonly modify the DNA at the start of the LINE-1 genes, which stops the LINE-1 proteins from being made. However, germline cells temporarily remove these DNA modifications at certain stages of development, and previous work in mice suggests that this is when LINE-1 moves.

When mouse germline cells remove DNA modifications, they activate a gene called *Tex19.1*. This led MacLennan, García-Cañadas et al. to ask whether this gene plays a role in regulating LINE-1 activity in germline cells. When mice were genetically engineered to inactivate the *Tex19.1* gene in developing sperm cells, levels of one of the LINE-1 proteins, called L1-ORF1p, increased. This indicates that *Tex19.1* most likely acts to keep the levels of this protein down.

To find out how *Tex19.1* does this, a technique called immunoprecipitation was used to pull the the protein encoded by the *Tex19.1* gene out of mouse cells to see which other proteins came along with it. The interacting proteins included L1-ORF1p and components of a molecular machine that identifies and marks undesired proteins for destruction. Furthermore, the levels of L1-ORF1p in mouse cells increased when this molecular machine (which is known as the ubiquitin system) was blocked. This suggests that cells use *Tex19.1* to keep LINE-1 in check by detecting its proteins and promoting their destruction.

The findings reveal that germline cells have another layer of defence that kicks in when DNA modifications are removed during development. In this situation, LINE-1 proteins are detected and destroyed before they can copy and paste the retrotransposon. Since LINE-1 retrotransposons have the potential to cause mutations in around one in every twenty people, if these findings are transferrable to humans, they could open new avenues for research into inherited mutations.

two proteins strictly required for retrotransposition (*Moran et al., 1996*): ORF1p is an RNA binding protein with nucleic acid chaperone activity (*Martin and Bushman, 2001*; *Hohjoh and Singer, 1997*), and ORF2p is a multidomain protein with reverse transcriptase and endonuclease activities (*Feng et al., 1996*; *Mathias et al., 1991*). Both these proteins interact directly or indirectly with various cellular factors and are incorporated into ribonucleoprotein particles (RNPs) along with the L1 RNA (*Beck et al., 2011*; *Goodier et al., 2013*; *Hancks and Kazazian, 2016*; *Richardson et al., 2014a*; *Taylor et al., 2013*). While these proteins exhibit a strong *cis*-preference to bind to and catalyse mobilization of their encoding mRNA, they can act in trans on other RNAs, including those encoded by SINEs (*Kulpa and Moran, 2006*; *Wei et al., 2001*; *Dewannieux et al., 2003*; *Esnault et al., 2000*). Some human L1s also encode a *trans*-acting protein, ORF0, that stimulates retrotransposition, although its mechanism of action is currently poorly understood (*Denli et al., 2015*). Host restriction mechanisms that regulate the activity of these L1-encoded proteins will impact on the stability of mammalian genomes and the incidence of genetic disease.

Regulating retrotransposon activity is particularly important in the germline as *de novo* retrotransposon integrations that arise in these cells can be transmitted to the next generation (*Crichton et al., 2014*). The mammalian germline encompasses lineage-restricted germ cells including primordial germ cells, oocytes, and sperm, and their pluripotent precursors in early embryos (*Ollinger et al., 2010*). L1 mobilization may be more prevalent in pluripotent cells in pre-implantation embryos rather than in lineage-restricted germ cells (*Kano et al., 2009*; *Richardson et al., 2017*), and regulation of L1 activity in the pluripotent phase of the germline cycle is therefore likely to have a significant effect on trans-generational genome stability. Repressive histone modifications

and DNA methylation typically suppress transcription of retrotransposons in somatic mammalian cells (*Beck et al., 2011*; *Hancks and Kazazian, 2016*; *Richardson et al., 2014a*; *Crichton et al., 2014*), but many of these transcriptionally repressive marks are globally removed during pre-implantation development and during fetal germ cell development in mice (*Hajkova et al., 2008*; *Popp et al., 2010*; *Santos et al., 2002*; *Fadloun et al., 2013*). DNA methylation in particular plays a key role in transcriptionally repressing L1 in the germline (*Bourc'his and Bestor, 2004*), and it is not clear how L1 activity is controlled in pluripotent cells and fetal germ cells while they are DNA hypomethylated. However, evidence suggests that L1 mobilization is tightly controlled in pluripotent cells to reduce trans-generational genome instability (*Wissing et al., 2012*; *Marchetto et al., 2013*).

In fetal germ cells, loss of DNA methylation correlates with relaxed transcriptional suppression of retrotransposons (*Molaro et al., 2014*), but also induces expression of methylation-sensitive germline genome-defence genes that have roles in post-transcriptionally repressing these elements (*Hackett et al., 2012*). The methylation-sensitive germline genome-defence genes include components of the PIWI-piRNA pathway. This pathway promotes *de novo* DNA methylation of retrotransposons in male germ cells, cleaves retrotransposon RNAs, and may also interfere with retrotransposon translation (*Fu and Wang, 2014*; *Xu et al., 2008*). However, while mice carrying mutations in the PIWI-piRNA pathway can strongly de-repress L1-encoded RNA and protein during spermatogenesis (*Aravin et al., 2007*; *Carmell et al., 2007*), increased L1 mobilization has not yet been reported in these mutant models. Indeed, the level of L1 expression at different stages of the germline cycle does not completely correlate with the ability of L1 to mobilize, and post-translational control mechanisms have been proposed to restrict the ability of L1 to mobilize in the mouse germline (*Kano et al., 2009*). However, the molecular identities of these post-translational L1 restriction mechanisms have not yet been elucidated.

We have previously shown that programmed DNA hypomethylation in the developing mouse germline induces expression of a group of genes that are involved in suppressing retrotransposon activity (*Hackett et al., 2012*). One of the retrotransposon defence genes induced in response to programmed DNA hypomethylation, *Tex19.1*, suppresses specific retrotransposon transcripts in spermatocytes (*Ollinger et al., 2008*; *Reichmann et al., 2012*), however its direct mechanism of action remains unclear. *Tex19.1* is expressed in germ cells, pluripotent cells and the placenta and is one of two *TEX19* orthologs generated by a rodent-specific gene duplication (*Kuntz et al., 2008*; *Wang et al., 2001*; *Ollinger et al., 2008*). These mammal-specific proteins have no functionally characterized protein motifs or reported biochemical activity, but mouse TEX19.1 is predominantly cytoplasmic in the germline (*Ollinger et al., 2008*; *Yang et al., 2010*). Here we show that *Tex19.1* regulates L1-ORF1p levels and mobilization of engineered L1 elements. We show that mouse TEX19.1, and its human ortholog TEX19, physically interact with L1-ORF1p, and regulate L1-ORF1p abundance through stimulating its polyubiquitylation and proteasome-dependent degradation. We show that TEX19.1 likely controls L1-ORF1p abundance in concert with UBR2, an E3 ubiquitin ligase that we show also physically interacts with and regulates L1-ORF1p levels *in vivo*. We also show that loss of *Tex19.1* results in increased L1-ORF1p abundance and increased mobilization of engineered L1 constructs in pluripotent mouse embryonic stem cells, suggesting that *Tex19.1* functions as a post-translational control mechanism to restrict L1 mobilization in the developing germline.

## Results

### L1-ORF1p abundance is post-transcriptionally regulated by *Tex19.1* in mouse germ cells

Programmed DNA hypomethylation in the developing germline induces expression of *Tex19.1*, which encodes a predominantly cytoplasmic protein in spermatocytes that suppresses retrotransposon expression through unknown mechanisms (*Ollinger et al., 2008*; *Reichmann et al., 2012*; *Yang et al., 2010*). In order to define the role of TEX19.1 in retrotransposon regulation in more detail we investigated whether *Tex19.1* might have post-transcriptional effects on cytoplasmic stages of the retrotransposon life cycle. Since *Tex19.1*$^{-/-}$ spermatocytes have defects in meiosis that induce spermatocyte death (*Ollinger et al., 2008*), we analysed mouse L1 ORF1p (mL1-ORF1p) expression in prepubertal testes during the first wave of spermatogenesis before any increased spermatocyte death is evident (*Ollinger et al., 2008*). Western blotting showed that P16 *Tex19.1*$^{-/-}$

testes have elevated levels of mL1-ORF1p (*Figure 1A*), even though L1 RNA levels do not change (*Figure 1B*), as previously shown (*Ollinger et al., 2008*; *Reichmann et al., 2012*). Primers designed against the active A, Gf and Tf subtypes of L1 (*de la Rica et al., 2016*) similarly did not detect any change in L1 RNA abundance in P16 *Tex19.1*$^{-/-}$ testes (*Figure 1—figure supplement 1A*). These data suggest that *Tex19.1* negatively regulates mL1-ORF1p post-transcriptionally in male germ cells. Immunostaining of P16 testes showed that, consistent with previous reports, mL1-ORF1p is expressed in meiotic spermatocytes in control mice (*Figure 1C*) (*Soper et al., 2008*; *Branciforte and Martin, 1994*). However, mL1-ORF1p immunostaining is elevated approximately two fold in the same cell type in *Tex19.1*$^{-/-}$ mice (*Figure 1C*). Thus, distinct from its role in transcriptional regulation of retrotransposons (*Ollinger et al., 2008*; *Reichmann et al., 2012*; *Crichton et al., 2017a*; *Reichmann et al., 2013*), *Tex19.1* appears to have a role in post-transcriptionally suppressing mL1-ORF1p abundance in meiotic spermatocytes.

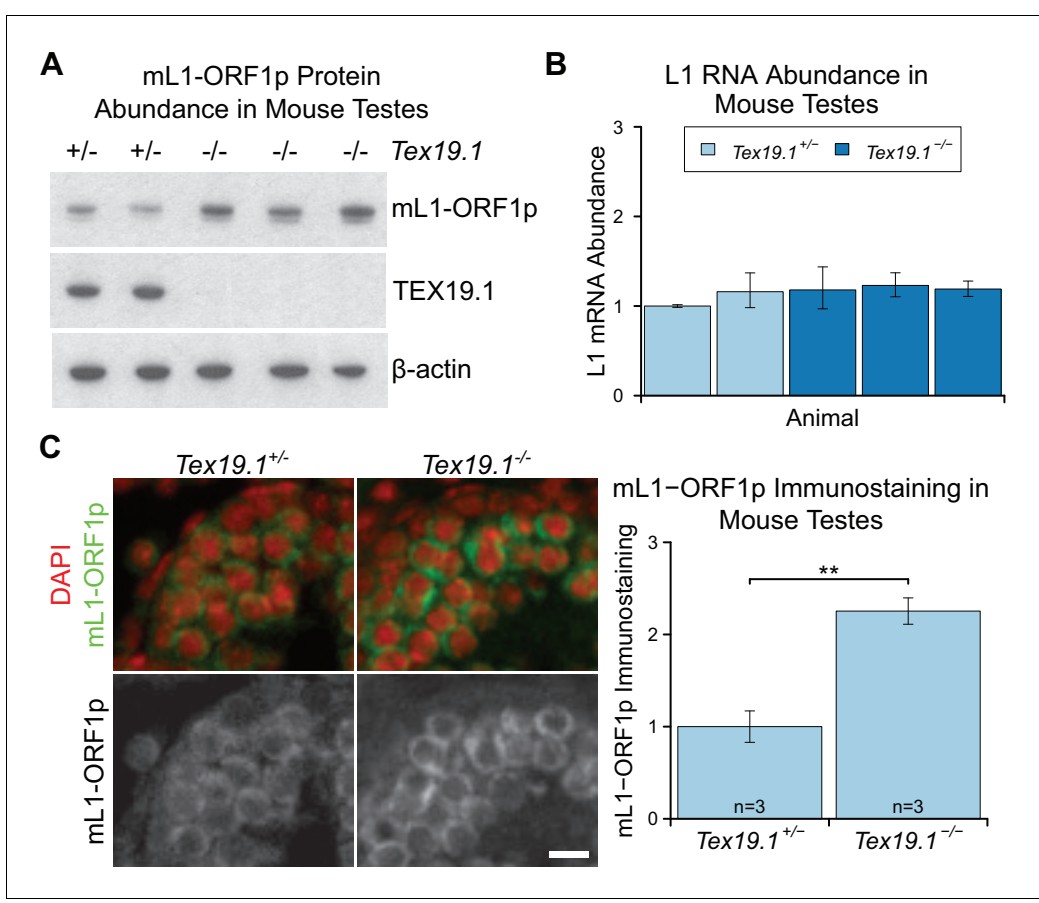

**Figure 1.** mL1-ORF1p is post-transcriptionally regulated by *Tex19.1* in mouse germ cells. (**A**) Western blot for mL1-ORF1p in *Tex19.1*$^{+/-}$ and *Tex19.1*$^{-/-}$ littermate P16 mouse testes. *β*-actin is a loading control. Data shown is representative of seven *Tex19.1*$^{-/-}$ animals across four litters. (**B**) qRT-PCR for L1 RNA using primers against ORF2 in testes from the same animals analyzed in panel A. Expression relative to *β*-actin was normalized to a *Tex19.1*$^{+/-}$ control animal. Error bars indicate SEM for three qPCR technical replicates from the same reverse-transcribed RNA. (**C**) Immunostaining for mL1-ORF1p (green) in *Tex19.1*$^{+/-}$ and *Tex19.1*$^{-/-}$ P16 mouse testis sections. Nuclei are counterstained with DAPI (shown as red). Scale bar, 10 μm. Anti-mL1-ORF1p immunostaining per unit area was quantified for three animals for each genotype, and normalized to the mean for *Tex19.1*$^{+/-}$ animals. Means ± SEM (1 ± 0.17 and 2.25 ± 0.14 for *Tex19.1*$^{+/-}$ and *Tex19.1*$^{-/-}$ respectively) are indicated; **p<0.01 (*t*-test, p=0.005).

The following figure supplement is available for figure 1:

**Figure supplement 1.** *Tex19.1* does not inhibit L1 translation.

## TEX19.1 interacts with multiple components of the ubiquitin-proteasome system

Post-transcriptional control of protein abundance can occur through regulation of mRNA translation or protein stability. To investigate whether TEX19.1 might be involved in one of these processes we attempted to identify RNAs or proteins that interact with TEX19.1. In contrast to the PIWI proteins MILI and MIWI (*Grivna et al., 2006*; *Unhavaithaya et al., 2009*), oligo(dT) pull-downs from mouse testicular lysate suggest that TEX19.1 is not physically associated with RNA in this tissue (*Figure 1—figure supplement 1B*) and neither is TEX19.1 enriched in testicular polysome fractions containing actively translating mRNAs (*Figure 1—figure supplement 1C*). In addition, the increase in mL1-ORF1p abundance in $Tex19.1^{-/-}$ testes is not accompanied by an increase in L1 RNA abundance in polysomes (*Figure 1—figure supplement 1D*). Therefore the increase in mL1-ORF1p abundance in $Tex19.1^{-/-}$ testes does not appear to reflect a direct role for TEX19.1 in regulating translation of L1 RNAs.

We next attempted to identify TEX19.1-interacting proteins in order to determine how TEX19.1 might regulate L1-ORF1p levels. TEX19.1 is endogenously expressed in mouse embryonic stem cells (ESCs) (*Kuntz et al., 2008*), and mass spectrometry of TEX19.1-YFP immunoprecipitates (IPs) from stably expressing mouse ESCs revealed co-IP of multiple components of the ubiquitin-proteasome system (*Figure 2A*, *Figure 2B*, *Supplementary file 1*, *Supplementary file 2*). TEX19.1-YFP IPs contained a strong co-immunoprecipitating band of approximately stoichiometric abundance to TEX19.1-YFP which was identified as UBR2, a RING domain E3 ubiquitin ligase and known interacting partner for TEX19.1 (*Yang et al., 2010*) (*Figure 2A*, *Figure 2B*, *Figure 2—figure supplement 1A*, *Figure 2—figure supplement 1B*). The identification of the only known interacting partner for TEX19.1 in this co-IP suggests that the TEX19.1-YFP construct used in this experiment recapitulates interactions relevant for endogenous TEX19.1. Indeed, all detectable endogenous TEX19.1 in ESCs co-fractionates with UBR2 in size exclusion chromatography (*Figure 2C*), consistent with TEX19.1 existing in a stable heteromeric complex with UBR2 in these cells. Importantly, *Ubr2* has previously been shown to be required for TEX19.1 protein stability in mouse testes (*Yang et al., 2010*) which, in combination with the co-fractionation and stoichiometric abundance of these proteins in the ESC IPs, suggests that any TEX19.1 protein not associated with UBR2 may be unstable and degraded. TEX19.1-YFP also co-IPs with additional components of the ubiquitin-proteasome system including UBE2A/B, an E2 ubiquitin-conjugating enzyme and cognate partner of UBR2 (*Kwon et al., 2003*; *Xie and Varshavsky, 1999*), and a HECT-domain E3 ubiquitin ligase, HUWE1 (*Chen et al., 2005*; *Liu et al., 2005*) (*Figure 2B*, *Supplementary file 2*). The physical associations between TEX19.1 and multiple components of the ubiquitin-proteasome system strongly suggest that the post-transcriptional increase in mL1-ORF1p abundance in $Tex19.1^{-/-}$ testes might reflect a role for TEX19.1 in regulating degradation of mL1-ORF1p.

## TEX19.1 orthologs directly interact with L1-ORF1p

We next tested if TEX19.1 might also interact with mL1-ORF1p. Although we did not identify any mL1-ORF1p peptides in the mass spectrometry analysis of TEX19.1-YFP IPs from ESCs, we did identify a single hL1-ORF1p peptide in similar IPs from stable TEX19.1-YFP expressing HEK293T cells (*Reichmann et al., 2017*). Since interactions between E3 ubiquitin ligases and their substrates are expected to be transient and weakly represented in IP experiments, we tested directly whether TEX19.1-YFP and epitope-tagged mL1-ORF1p interact by co-expressing these proteins in HEK293T cells and immunoprecipitating either TEX19.1-YFP or epitope-tagged mL1-ORF1p. Both IPs revealed weak reciprocal interactions between TEX19.1-YFP and T7 epitope-tagged mL1-ORF1p (mL1-ORF1p-T7) (*Figure 2D*, *Figure 2—figure supplement 1C*). Although human TEX19 is significantly truncated relative to its mouse ortholog, the physical interaction between TEX19 and L1-ORF1p is conserved in humans (*Figure 2—figure supplement 1D*, *Figure 2—figure supplement 1E*).

We next tested whether the biochemical interaction between TEX19.1-YFP and mL1-ORF1p-T7 is reflected by co-localization of these proteins. TEX19.1 is predominantly cytoplasmic in ES cells and in germ cells (*Ollinger et al., 2008*; *Yang et al., 2010*), but in the hypomethylated placenta and when expressed in somatic cell lines, TEX19.1 can localize to the nucleus (*Kuntz et al., 2008*; *Reichmann et al., 2013*). The context-dependent localization of TEX19.1 suggests that TEX19.1-interacting proteins in ES cells and germ cells could retain this protein in the cytoplasm in these cell

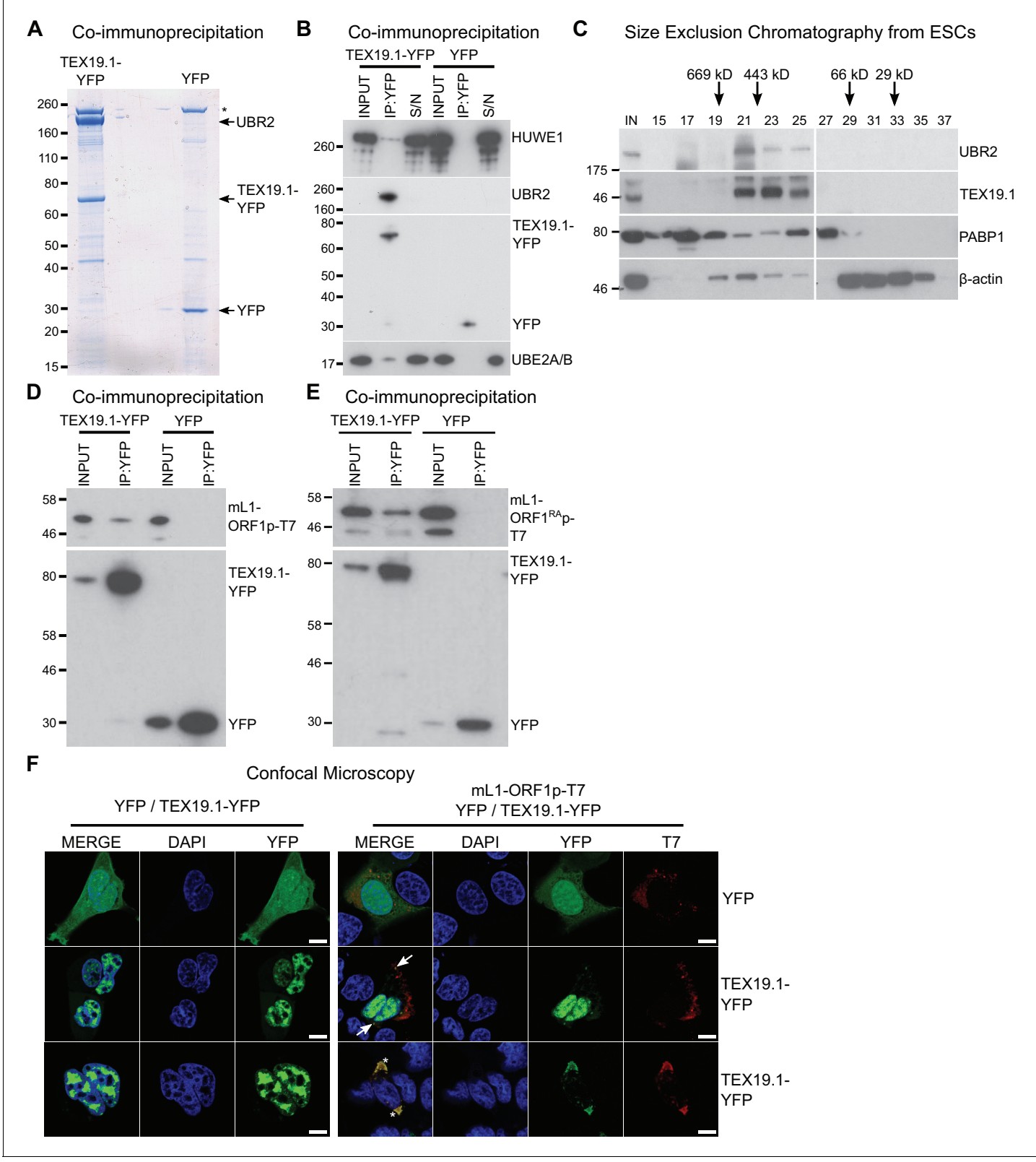

**Figure 2.** TEX19.1 physically interacts with components of the ubiquitin proteasome system and with L1-ORF1p. (**A**) Colloidal blue-stained cytoplasmic anti-YFP immunoprecipitates from mouse ESCs stably expressing mouse TEX19.1-YFP or YFP. Mass spectrometry identities of major bands are indicated, and a non-specific band marked with an asterisk. (**B**) Western blots for ubiquitin-proteasome system components in anti-YFP immunoprecipitates (IPs) from panel A. Anti-YFP IP inputs, IPs and IP supernatants (S/N) were blotted with indicated antibodies. (**C**) Size exclusion

*Figure 2 continued on next page*

*Figure 2 continued*

chromatography of cytoplasmic extract from ESCs showing elution of endogenous mouse TEX19.1 and UBR2. PABP1 and β-actin are included as controls. Input (IN) sample is also shown, and eluted fraction numbers and the positions of pre-stained molecular weight markers in kD are indicated. (D, E) IPs from HEK293T cells co-transfected with mL1-ORF1p-T7 constructs and either mouse TEX19.1-YFP or YFP and Western blotted with indicated antibodies. The mutant mL1-ORF1$^{RA}$p in panel E has a reduced binding affinity for RNA (*Kulpa and Moran, 2005*; *Martin et al., 2005*). (F) Subcellular localization of mouse TEX19.1-YFP in the presence and absence of mL1-ORF1p-T7. U2OS cells were transiently transfected with TEX19.1-YFP or YFP expression constructs with or without a plasmid expressing mL1-ORF1p-T7 (pCEPL1SM-T7), then stained with anti-T7 antibodies, and with DAPI to detect DNA. 49% of 51 cells examined exhibited some co-localization of mL1-ORF1p-T7 with TEX19.1-YFP. In 71% of these co-localizing cells mL1-ORF1p-T7 with TEX19.1-YFP were both present in a subset of small cytoplasmic foci (arrows). In the remaining 29% of co-localizing cells, large cytoplasmic aggregates of mL1-ORF1p-T7 extensively co-localize with TEX19.1-YFP (asterisks). Two representative images of cells transfected with either TEX19.1-YFP alone or TEX19.1-YFP in combination with mL1-ORF1p are shown. Scale bars 10 μm.

The following figure supplement is available for figure 2:

**Figure supplement 1.** TEX19 orthologs interact with UBR2 and L1-ORF1p.

types. L1-ORF1p has been reported to form cytoplasmic aggregates that co-localize with stress granule markers (*Doucet et al., 2010*; *Goodier et al., 2007*), therefore we tested whether co-expression of L1-ORF1p and TEX19.1 might localize TEX19.1 to these L1-ORF1p-containing aggregates. As expected, confocal microscopy showed that TEX19.1-YFP localizes to the nucleus when expressed in U2OS cells, however co-expression with mL1-ORF1p-T7 resulted in some co-localization of both these proteins in cytoplasmic aggregates in 25 of 51 cells examined. In 71% of these co-localizing cells, TEX19.1 and mL1-ORF1p-T7 exhibited partial co-localization in some cytoplasmic aggregates (*Figure 2F*). In the remaining 29% co-localizing cells, more extreme co-localization was observed with expression of mL1-ORF1p-T7 re-localizing all detectable TEX19.1-YFP out of the nucleus and into cytoplasmic aggregates (*Figure 2F*). In sum, these co-localization data are consistent with the co-IP data suggesting that TEX19.1-YFP and mL1-ORF1p-T7 physically interact, likely in a transient manner.

A number of host factors have been shown to associate with L1-ORF1p, although many of these interactions are indirect and mediated by RNA, likely reflecting interactions within the L1 RNP (*Goodier et al., 2013*; *Taylor et al., 2013*; *Moldovan and Moran, 2015*). However, the interaction between host PCNA and L1-ORF2p is resistant to RNase treatment and is therefore a good candidate to be a direct interaction (*Taylor et al., 2013*). We therefore tested whether the interaction between TEX19.1 and L1-ORF1p might be direct and independent of RNA. TEX19.1-YFP is able to interact with a mutant allele of mL1-ORF1p which has severely impaired binding to RNA and impaired L1 mobilization (*Kulpa and Moran, 2005*; *Martin et al., 2005*) (*Figure 2E*, *Figure 2—figure supplement 1F*), suggesting that the interaction between TEX19.1-YFP and mL1-ORF1p is RNA-independent and could potentially be direct. We next tested whether bacterially expressed human TEX19 might interact with bacterially expressed hL1-ORF1p. Notably, co-expression of double-tagged human MBP-TEX19-GB1-His$_6$ with Strep-tagged human L1-ORF1p (Strep-hL1-ORF1p) in bacteria resulted in a strong interaction between these proteins, and isolation of a stable TEX19-hL1-ORF1p complex (*Figure 3A*, *Figure 3B*). This interaction required the proteins to be co-expressed (*Figure 3A*) and was resistant to micrococcal nuclease treatment (*Figure 3B*). Furthermore, TEX19 was found to recognize the conserved and previously crystallized part of the hL1-ORF1p trimer (*Khazina et al., 2011*; *Boissinot and Sookdeo, 2016*) and the N-terminal half of hL1-ORF1p that lacks the RNA-binding domains (*Figure 3C*, *Figure 3E*). In addition, the first 68 amino acids of TEX19, which contain the conserved MCP region and a putative Zn$^{2+}$-binding motif (*Bianchetti et al., 2015*) were found to be necessary and sufficient for the interaction (*Figure 3D*, *Figure 3F*). Consequently, the MCP region of TEX19 might contact the conserved C-terminal half of the coiled-coil domain, which is present in both L1-ORF1p fragments tested for interactions, although additional contacts between the variable parts of the two proteins can not be excluded. Taken together, the co-IPs, the co-localization and the isolation of a TEX19:L1-ORF1p complex from bacterially expressed proteins suggest that TEX19 directly interacts with L1-ORF1p in a conserved manner and, to our knowledge, represents the first example of a host protein that directly binds to the retrotransposon-encoded protein L1-ORF1p from mammals.

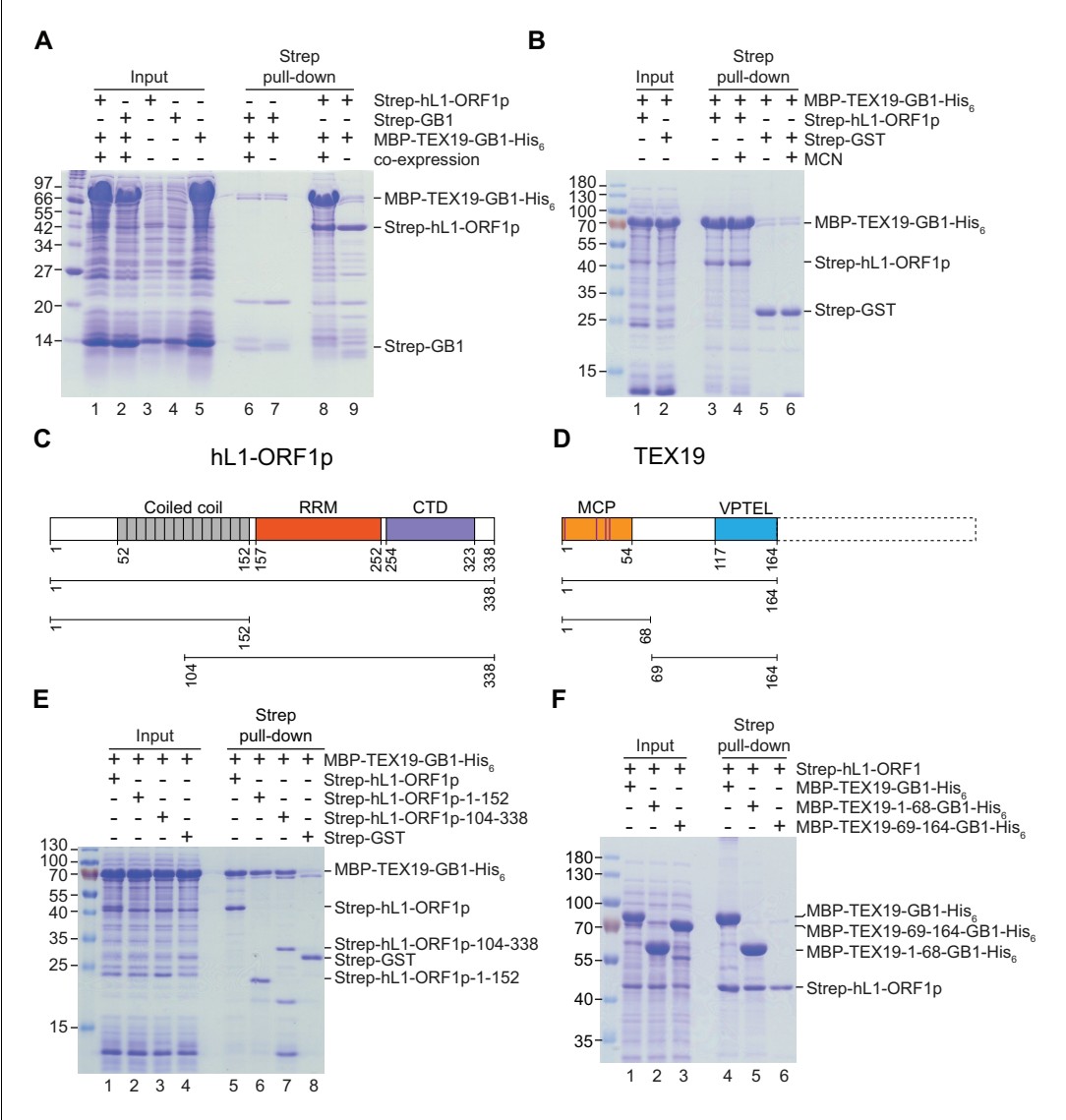

**Figure 3.** Direct interaction between human TEX19 and human L1-ORF1p. (**A**) Strep pull-down assays from bacterial (*Escherichia coli*) lysates. Double-tagged human TEX19 was either co-expressed with Strep-tagged human L1-ORF1p (lane 8) or added after L1-ORF1p immobilization on Strep-Tactin beads (lane 9). Strep-GB1 served as a control (lanes 6 and 7). (**B**) Pull-down assays of the co-expressed proteins in the absence and presence of micrococcal nuclease (MCN, lanes 3 and 4). Strep-GST served as a control (lanes 5 and 6). (**C**) Bar diagram of human L1-ORF1p based on the crystal structure by *Khazina et al. (2011)* and consistent with the alignment by *Boissinot and Sookdeo (2016)*. Structural domains are colored and the sub-fragments used for pulldown assays are indicated below the bar with the corresponding amino acid numbers. The C-terminal fragment is sufficient for L1-ORF1p trimerization and has been crystallized. The N-terminal fragment is highly variable among mammals. (**D**) Bar diagram of human TEX19 according to the alignment by *Bianchetti et al. (2015)*. The conserved MCP and VPTEL regions are colored and the C-terminal extension that is present in murine TEX19.1 and most of the other mammalian homologs is indicated with a dotted line. Purple lines indicate a putative CHCC zinc-binding motif in the MCP region. (**E**) Strep pull-down assays with bacterially expressed sub-fragments of human L1-ORF1p and full-length human TEX19. (**F**) Strep pull-down assays with bacterially expressed sub-fragments of human TEX19 and full-length human L1-ORF1p.

## *Tex19.1* orthologs stimulate polyubiquitylation and degradation of L1-ORF1p

The strong interaction between TEX19 and hL1-ORF1p seen with bacterially-expressed proteins contrasts with weaker interactions detected in HEK293T cells. However, it is possible that the difference in the strength of these interactions reflects the presence of UBR2 in mammalian cells, which allows

a TEX19-UBR2 complex to assemble and transiently interact with hL1-ORF1p to catalyse its ubiquity-lation and subsequent degradation. We therefore investigated if L1-ORF1p is ubiquitylated and degraded by the proteasome, and whether this might be stimulated by TEX19. Endogenously expressed mL1-ORF1p in mouse testes represents a collection of protein molecules expressed from hundreds of variant copies of L1 at different genomic loci (*Chinwalla et al. 2002*). Therefore, to allow us to correlate the abundance of L1-ORF1p with its encoding RNA more accurately, and to detect transient polyubiquitylated intermediates that are destined for proteasome-dependent deg-radation, we expressed engineered epitope-tagged hL1-ORF1p constructs in HEK293T cells. HEK293T cells do not endogenously express detectable levels of *TEX19* (*Reichmann et al., 2017*) and cell-based ubiquitylation assays show that there is basal ubiquitylation of hL1-ORF1p in these cells, detectable as a ladder of hL1-ORF1p species in $his_6$-myc-Ub pull-downs (*Figure 4A*). The increasing molecular weights of these bands presumably correspond to increasing ubiquitylation of hL1-ORF1p. Furthermore, treating these cells with the proteasome inhibitor MG132 showed that hL1-ORF1p abundance is negatively regulated by the proteasome in the absence of *TEX19* expres-sion (*Figure 4B*). Interestingly, co-expression of *TEX19* during the cell-based ubiquitylation assay increases polyubiquitylation of hL1-ORF1p (*Figure 4C*, *Figure 4—figure supplement 1A*). *TEX19* expression increases the proportion of hL1-ORF1p-T7 that has at least four ubiquitin monomers, the minimum length of polyubiquitin chain required to target proteins to the proteasome (*Thrower et al., 2000*). These cell-based ubiquitylation assays were performed in the absence of proteasome inhibitor as this treatment can cause the TEX19.1-interacting protein UBR2, and poten-tially also other regulators of L1-ORF1p, to accumulate (*An et al., 2012*). Therefore, we cannot determine whether *TEX19* also influences additional more extensively polyubiquitylated species of hL1-ORF1p that are more rapidly degraded by the proteasome. Nevertheless, expression of *TEX19* in these cells is sufficient to reduce the abundance of the T7-tagged hL1-ORF1p protein without any change in the abundance of its encoding RNA (*Figure 4D*). The ability of *TEX19* to regulate L1-ORF1p abundance is not restricted to HEK293T cells, and expression of either mouse or human *TEX19* orthologs reduces both mouse and human L1-ORF1p levels in hamster XR-1 cells (*Figure 4—figure supplement 1B*, *Figure 4—figure supplement 1C*). Taken together, these gain-of-function data for *TEX19* mirror the loss-of-function data obtained from $Tex19.1^{-/-}$ testes, confirm that the increased mL1-ORF1p levels in $Tex19.1^{-/-}$ testes are not a consequence of altered progression of $Tex19.1^{-/-}$ spermatocytes through meiosis (*Crichton et al., 2017b*; *Ollinger et al., 2008*), and strongly suggest that *Tex19.1* orthologs function to post-translationally regulate L1-ORF1p abun-dance. The ubiquitylation and interaction data together suggests that, *TEX19* orthologs regulate L1-ORF1p abundance by molecular recognition of L1-ORF1p and stimulation of its polyubiquitylation and proteasome-dependent degradation.

## *Tex19.1* orthologs restrict mobilization of engineered L1 constructs

L1-ORF1p has essential roles in L1 retrotransposition (*Beck et al., 2011*; *Richardson et al., 2014a*; *Hancks and Kazazian, 2016*) and is strictly required for the retrotransposition of engineered L1 con-structs in cultured mammalian cells (*Moran et al., 1996*). Since TEX19 orthologs bind to L1-ORF1p and negatively regulate its abundance, we next investigated whether *Tex19.1* might inhibit L1 mobi-lization in cultured cells. Engineered L1 retrotransposition assays with an EGFP retrotransposition indicator cassette (*Ostertag et al., 2000*; *Coufal et al., 2009*) (*Figure 5A*) were used to measure the effect of *Tex19.1* on the mobilization rate of active mouse L1 elements (*Goodier et al., 2001*; *Han and Boeke, 2004*) in HEK293T cells. Notably, expression of *Tex19.1* reduced the ability of both a codon-optimized $T_f$ type and a natural $G_f$ type mouse L1 to mobilize in these cells, suggesting that *Tex19.1* restricts retrotransposition of multiple active L1 subtypes (*Figure 5B*). Control experiments verified that a mouse L1 carrying missense mutations in the EN and RT domains of ORF2 (mouse $L1^{mut2}$) failed to retrotranspose in this assay (*Figure 5B*), and that retrotransposition was potently inhibited by the restriction factor APOBEC3A (*Bogerd et al., 2006a*; *Bogerd et al., 2006b*) (*Figure 5B*). Mouse *Tex19.1* also restricts mobilization of engineered human L1 constructs (*Fig-ure 5—figure supplement 1A*) although less efficiently than it restricts mouse L1s. Altogether, these data show that *Tex19.1* can function as a restriction factor for L1 mobilization in cultured cells.

Mouse *Tex19.1* expression is activated in response to DNA hypomethylation in multiple contexts (*Hackett et al., 2012*), and in humans *TEX19* is a cancer testis antigen expressed in multiple types of tumor where it is associated with poor cancer prognosis (*Feichtinger et al., 2012*; *Planells-*

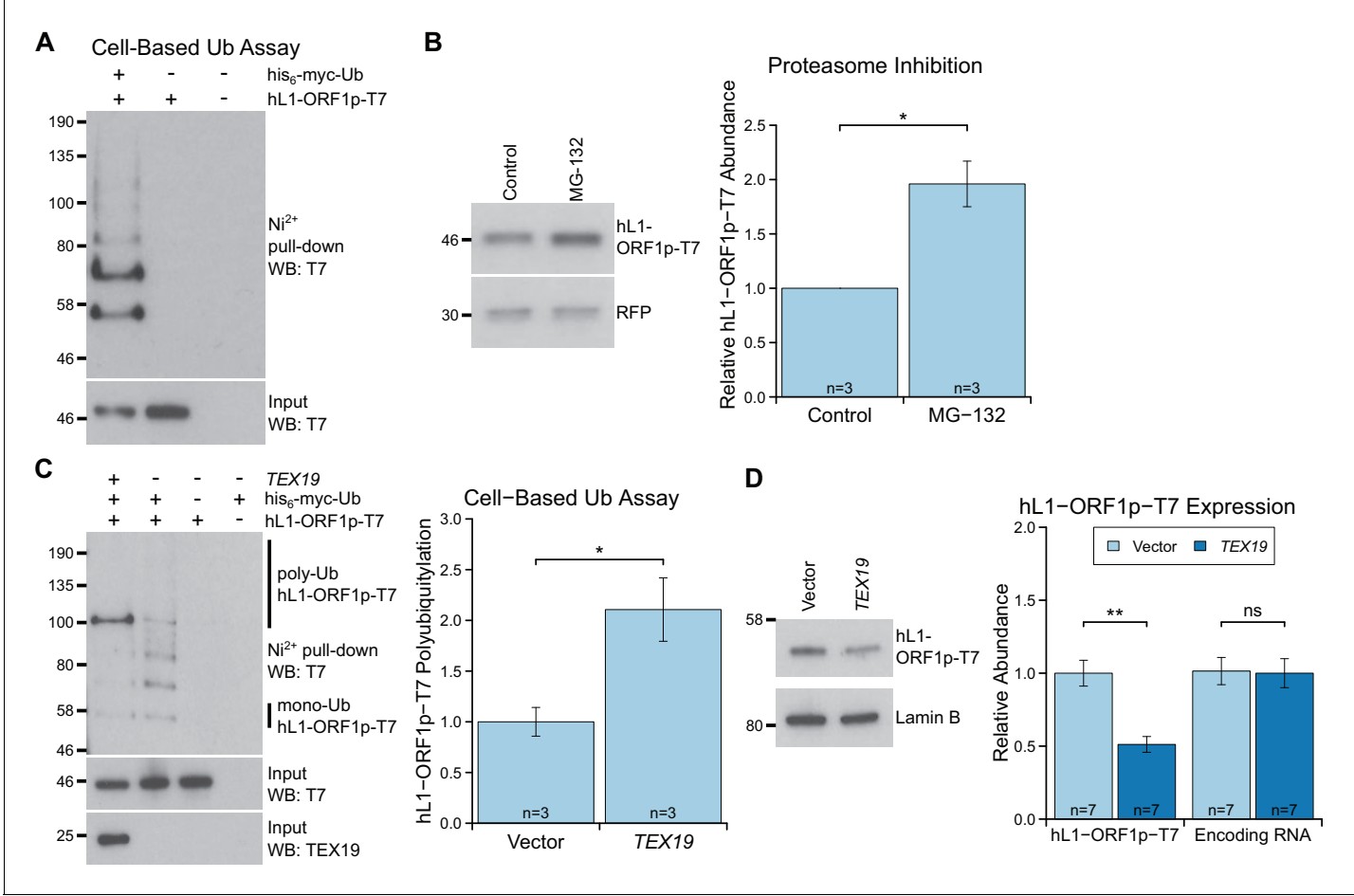

**Figure 4.** *TEX19* stimulates polyubiquitylation of hL1-ORF1p. (A) Cell-based ubiquitylation assay (Ub assay) for T7 epitope-tagged hL1-ORF1p in HEK293T cells. HEK293T cells were transfected with hL1-ORF1p-T7 and his$_6$-myc-ubiquitin (his$_6$-myc-Ub), and his$_6$-tagged proteins isolated using Ni$^{2+}$ agarose. Inputs and Ni$^{2+}$ pull-downs were analysed by Western blotting for T7. (B) Western blots and quantification of hL1-ORF1p-T7 abundance in HEK293T cells after treatment with either the proteasome inhibitor MG132 (50 µM, 7 hr) or DMSO as a vehicle control. HEK293T cells were co-transfected with hL1-ORF1p-T7 and RFP to control for transfection efficiency, and hL1-ORF1p-T7 abundance measured relative to RFP, then normalized to the DMSO controls for three independent transfections. MG132 treatment increases hL1-ORF1p-T7 abundance 1.96 ± 0.21 fold. *p<0.05 (*t*-test, p=0.04). (C) Cell-based ubiquitylation assay (Ub assay) for hL1-ORF1p-T7 in HEK293T cells in the presence and absence of human *TEX19*. Ni$^{2+}$-pull downs were Western blotted (WB) with anti-T7 antibodies. Polyubiquitylated hL1-ORF1p-T7 containing four or more ubiquitin molecules (~100 kD band and above) was quantified relative to monoubiquitylated hL1-ORF1p-T7 (~58 kD band) and normalized to empty vector controls. Means ± SEM (1 ± 0.14 and 2.11 ± 0.31 for vector control and *TEX19* respectively) are indicated; *p<0.05 (*t*-test, p=0.03). (D) Western blots of HEK293 FlpIn cells stably expressing hL1-ORF1p-T7 transfected with human *TEX19* or empty vector. Abundance of hL1-ORF1p-T7 protein and its encoding RNA were measured relative to lamin B and GAPDH respectively, and normalized to empty vector controls. Means ± SEM (1 ± 0.09 and 0.51 ± 0.06 for protein abundance and 1.01 ± 0.09 and 1 ± 0.10 for RNA abundance for vector control and *TEX19* respectively) are indicated; **p<0.01; ns indicates not significant (*t*-test, p=0.0005, 0.9 from left to right); Pre-stained MW markers (kD) are indicated beside blots.

The following figure supplement is available for figure 4:

**Figure supplement 1.** *TEX19* orthologs regulate L1-ORF1p abundance.

*Palop et al., 2017*). We therefore tested whether expression of *TEX19* orthologs might be sufficient to restrict L1 mobilization in multiple host cell types. L1 retrotransposition assays using a blasticidin retrotransposition indicator cassette (*Beck et al., 2010*; *Goodier et al., 2007*; *Morrish et al., 2002*) in HeLa cells (*Figure 5C*) showed that mouse *Tex19.1* similarly restricts mobilization of mouse and human L1 constructs by ~50% in this epithelial carcinoma cell line (*Figure 5D*). Human *TEX19* also restricts mobilization of mouse and human L1 constructs by ~ 50% in HeLa cells (*Figure 5D*). Similar

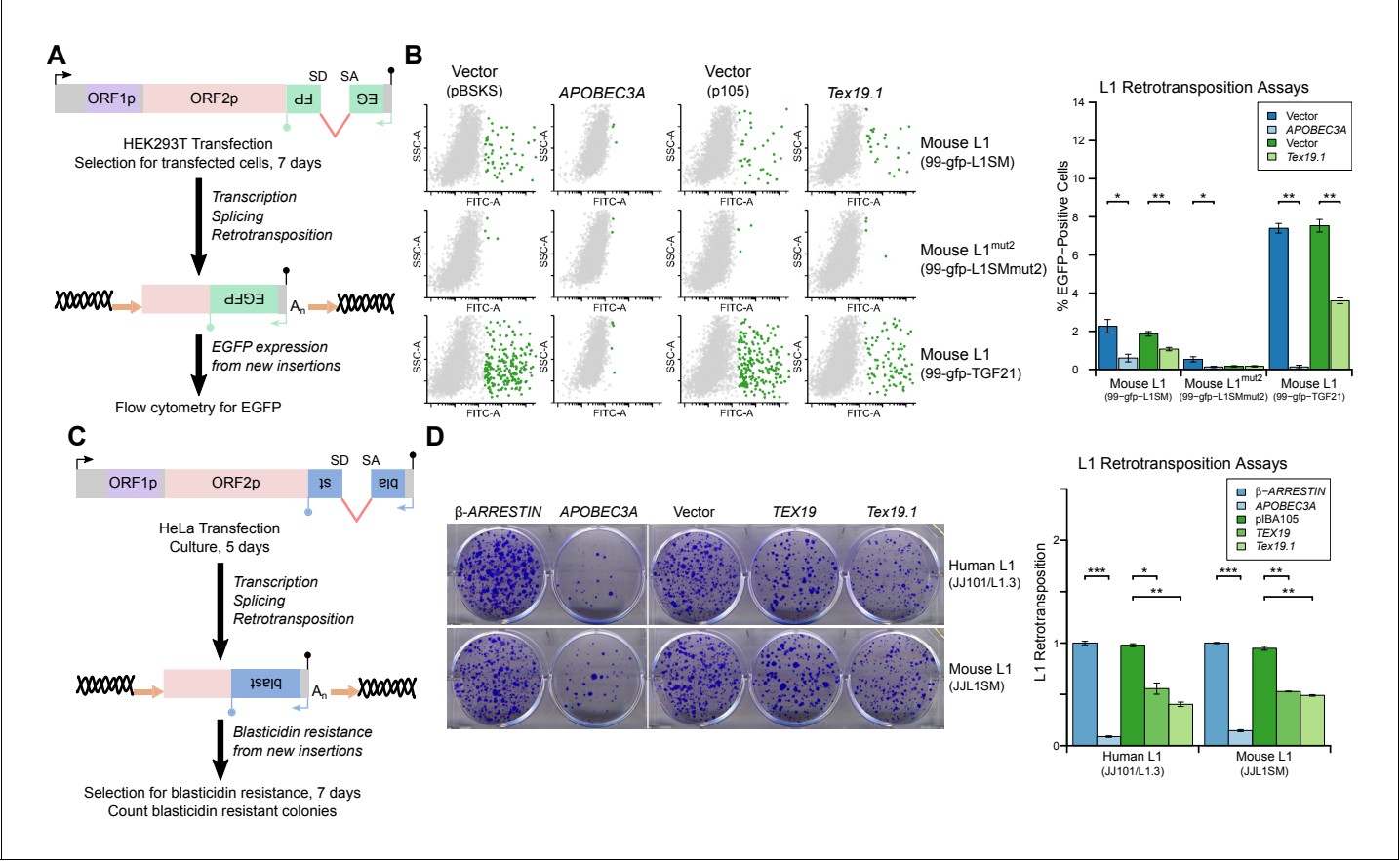

**Figure 5.** *TEX19* orthologs restrict L1 mobilization. (**A**) Schematic of engineered L1 retrotransposition assay in HEK293T cells using an EGFP indicator cassette. (**B**) Flow cytometry profiles from engineered mouse L1 retrotransposition assays performed as shown in panel A. HEK293T cells were co-transfected with engineered mouse L1 retrotransposition constructs containing EGFP indicator cassettes (99-gfp-L1SM, 99-gfp-L1SMmut2, 99-gfp-TGF21), and either Strep-tagged mouse *Tex19.1*, *APOBEC3A* (positive control) or empty vectors (pBSKS for *APOBEC3A*, pIBA105 for *Tex19.1*). EGFP fluorescence is plotted on the x-axis and side scatter on the y-axis of the flow cytometry profiles, and cells classed as EGFP-positive are shown in green. 99-gfp-L1SMmut2 carries missense mutations in the endonuclease and reverse transcriptase domains of ORF2p. *p<0.05; **p<0.01 (t-test, p=0.04, 0.006, 0.04, 1, 0.00001, 0.0004 for each pairwise comparison with vector from left to right). (**C**) Schematic of engineered L1 retrotransposition assays in HeLa cells using a blasticidin resistance indicator cassette. (**D**) Plates stained with 0.1% crystal violet showing blasticidin-resistant colonies from engineered L1 retrotransposition assays performed as shown in panel C. Human (JJ101/L1.3) and mouse (JJL1SM) L1 retrotransposition constructs containing blasticidin resistance indicator cassettes were co-transfected with *β-ARRESTIN* or *APOBEC3A* as negative and positive controls respectively, or with Strep-tagged mouse *Tex19.1*, Strep-tagged human *TEX19* or pIBA105 empty vector. Quantification of L1 retrotransposition was calculated relative to the *β-ARRESTIN* control. *p<0.05; **p<0.01 (t-test, p=0.0004, 0.02, 0.002, 0.0002, 0.002, 0.002 for each pairwise comparison with vector from left to right).

The following figure supplement is available for figure 5:

**Figure supplement 1.** *TEX19* orthologs restrict L1 mobilization.

effects on mobilization of L1 constructs were also observed in U2OS osteosarcoma cells (*Figure 5— figure supplement 1B*, *Figure 5—figure supplement 1C*). Thus, *TEX19* orthologs are host restriction factors for L1 retrotransposition in mice and humans. Importantly, although we have also shown that *TEX19* orthologs promote polyubiquitylation and degradation of L1-ORF1p, since TEX19 can directly bind to L1-ORF1p it is possible that this interaction also disrupts aspects L1-ORF1p function and contributes to TEX19-dependent restriction of L1 mobilization. Moreover, there could be additional aspects of TEX19 function that may also be contributing to its ability to restrict L1 mobilization. Indeed, it is not uncommon for host restriction factors to influence multiple aspects of

retrotransposon or retroviral life cycles (*Wang et al., 2010*; *Burdick et al., 2010*; *Goodier et al., 2012*; *Holmes et al., 2007*).

## UBR2 interacts with L1-ORF1p and regulates L1 independently of *Tex19.1* orthologs

The stoichiometric abundance of TEX19.1 and UBR2 in co-IPs in combination with the co-fractionation of all detectable TEX19.1 protein with UBR2 (*Figure 2A*, *Figure 2C*) suggests that *TEX19*-dependent polyubiquitylation of L1-ORF1p, and possibly also TEX19-dependent restriction of L1 mobilization, might be mediated by UBR2. In contrast to *Tex19.1*, *Ubr2* is ubiquitously expressed (*Figure 6—figure supplement 1A*) and UBR2 could contribute to basal ubiquitylation of L1-ORF1p in HEK293T cells (*Figure 4A*) and other somatic cell types. Thus, TEX19.1 could simply stimulate this activity when transcriptionally activated by programmed DNA hypomethylation in the developing germline. A simple test of this model would be that TEX19.1-dependent effects on L1-ORF1p abundance or L1 mobilization ought to be abolished in a *Ubr2* mutant background. However, the requirement for UBR2 to stabilize TEX19.1 protein (*Yang et al., 2010*) confounds analysis of the downstream requirement of UBR2 catalytic activity in TEX19.1-dependent functions: as TEX19.1 protein is unstable and undetectable in the absence of UBR2 (*Yang et al., 2010*), TEX19.1 might be expected to be unable to stimulate L1-ORF1p degradation or restrict L1 mobilization regardless of whether the E3 ubiquitin ligase activity of UBR2 is required for these functions or not. Indeed, $Ubr2^{-/-}$ testes largely phenocopy $Tex19.1^{-/-}$ testes, including transcriptional de-repression of MMERVK10C LTR retrotransposons (*Crichton et al., 2017a*).

To dissociate the effects of UBR2 on stability of TEX19.1 protein from potential effects on L1-ORF1p abundance and L1 mobilization, we tested whether UBR2 can regulate L1 in the absence of effects on TEX19 stability by using somatic HEK293T cells. Interestingly, mouse UBR2 co-IPs with mL1-ORF1p in HEK293T cells (*Figure 6A*), a cell type that does not express any detectable TEX19 protein (*Reichmann et al., 2017*). Thus, these data strongly suggest that UBR2 is able to regulate L1-ORF1p independently of any effects on TEX19 protein stability. UBR2 also interacts with mL1-ORF1$^{RA}$p mutants that have reduced binding to RNA (*Figure 6B*), suggesting that this physical interaction is not mediated by L1 RNA. Furthermore, these interactions are conserved in human L1-ORF1p (*Figure 6—figure supplement 1B*, *Figure 6—figure supplement 1C*). In addition, overexpression of UBR2 alone restricts mobilization of an engineered human L1 (*Figure 6C*). Thus, at least in overexpression experiments, UBR2 is able to physically interact with L1-ORF1p and restrict mobilization of L1 constructs in cultured cells.

To investigate regulation of hL1-ORF1p abundance by UBR2 further, we generated *UBR2* mutant HEK293T cell lines by CRISPR/Cas9-mediated genome editing. However, these cell lines grew slowly and poorly in culture, presumably reflecting the normal cellular roles of UBR2 in cohesin regulation, DNA repair, and chromosome stability (*Ouyang et al., 2006*; *Reichmann et al., 2017*). Therefore, to allow a meaningful analysis of the role of endogenous UBR2 in L1 regulation we analysed $Ubr2^{-/-}$ mice (*Figure 6—figure supplement 1D*, *Figure 6—figure supplement 1E*) which, despite having defects in spermatogenesis and female lethality, are otherwise grossly normal (*Kwon et al., 2003*). Notably, mL1, but not *Tex19.1* (*Figure 6—figure supplement 2*), is expressed in the brain (*Wang et al., 2001*; *Muotri et al., 2010*), therefore we used this tissue to assess whether *Ubr2* might have a *Tex19.1*-independent role in regulating mL1-ORF1p. Consistent with the physical interaction between UBR2 and mL1-ORF1p (*Figure 6A*), we found that mL1-ORF1p abundance is post-transcriptionally elevated approximately four fold in the cerebellum of $Ubr2^{-/-}$ mice (*Figure 6D*), suggesting that UBR2 may directly regulate polyubiquitylation and subsequent degradation of mL1-ORF1p *in vivo*. Interestingly, loss of *Ubr2* has no detectable effect on mL1-ORF1p abundance in the cerebrum (*Figure 6—figure supplement 1E*), which may reflect cell type specific differences in L1 regulation or genetic redundancy between UBR-domain proteins (*Tasaki et al., 2005*). Nevertheless, regardless of this additional complexity in the cerebrum, the increased abundance of mL1-ORF1p in $Ubr2^{-/-}$ cerebellum demonstrates that endogenous *Ubr2* plays a *Tex19.1*-independent role in regulating mL1-ORF1p abundance *in vivo*. Ubr2 has numerous endogenous cellular substrates and host functions beyond regulating mL1-ORF1p (*Ouyang et al., 2006*; *Reichmann et al., 2017*; *Sriram et al., 2011*), but expression of *Tex19.1* in the germline or in response to DNA hypomethylation appears to stimulate a pre-existing activity of UBR2 to regulate mL1-ORF1p, possibly at the expense of UBR2's activity towards some endogenous cellular substrates (*Reichmann et al., 2017*).

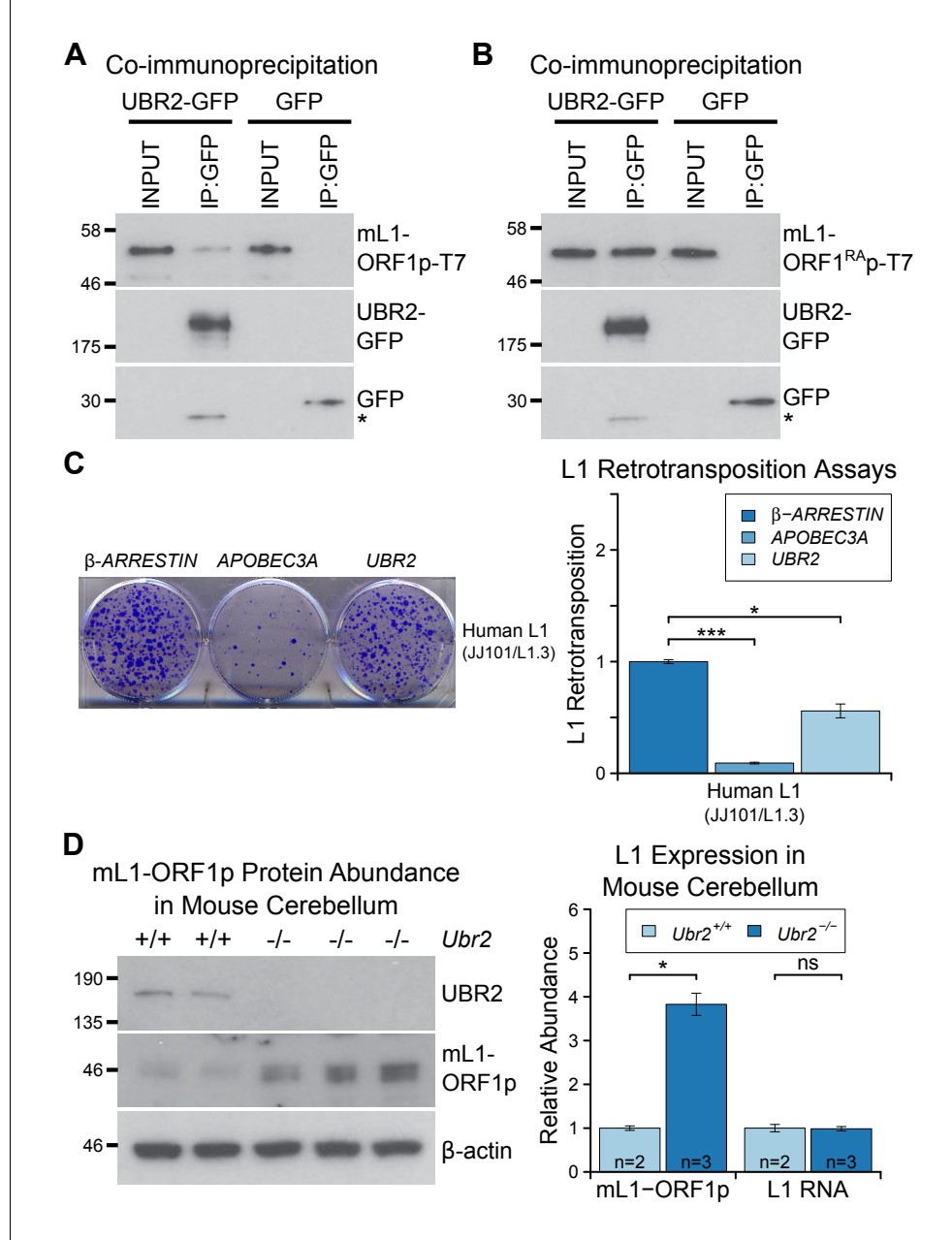

**Figure 6.** The TEX19.1-interacting protein UBR2 negatively regulates mL1-ORF1p abundance and L1 mobilization. (**A**) Co-immunoprecipitations (co-IPs) from HEK293T cells co-transfected with mL1-ORF1p-T7 and either mouse UBR2-GFP or GFP. IP inputs and IPs were Western blotted with T7 and GFP antibodies. A presumed cleavage product of UBR2-GFP running smaller than GFP itself is indicated with an asterisk. (**C**) Plates from an engineered L1 retrotransposition assay as described in *Figure 5C* stained with 0.1% crystal violet showing blasticidin-resistant colonies. Human (JJ101/L1.3) L1 retrotransposition construct was co-transfected with *β-ARRESTIN* or *APOBEC3A* as negative and positive controls respectively, or with UBR2-Flag. *p<0.05; ***p<0.01 (*t*-test, p=0.0004, 0.02 from left to right). (**D**) Western blots of endogenous UBR2 and mL1-ORF1p in P16 *Ubr2$^{+/+}$* and *Ubr2$^{-/-}$* mouse cerebellum. *β*-actin was used as a loading control. Quantification of mL1-ORF1p-T7 and L1 mRNA relative to *β*-actin and normalized to *Ubr2$^{+/+}$* control mice is also shown. Means ± SEM are indicated (1 ± 0.05 and 3.82 ± 0.25 for *Ubr2$^{+/+}$* and *Ubr2$^{-/-}$* respectively) *p<0.05; ns indicates not significant (*t*-test, p=0.048, 0.9 from left to right); pre-stained MW markers (kD) are shown beside blots.

The following figure supplements are available for figure 6:

*Figure 6 continued on next page*

*Figure 6 continued*

**Figure supplement 1.** The ubiquitously-expressed E3 ubiquitin ligase UBR2 physically interacts with L1-ORF1p but does not regulate its abundance in the cerebrum.

**Figure supplement 2.** *Tex19.1* expression is not detectable in brain.

### *Tex19.1* regulates mL1-ORF1p abundance and restricts L1 mobilization in pluripotent cells

As outlined earlier, L1 mobilization is thought to occur primarily in pluripotent cells within the germline cycle (*Kano et al., 2009*; *Richardson et al., 2017*), and regulation of L1 expression and mobilization in these cells is likely to significantly impact on the ability of L1 to influence germline mutation and genome evolution. Therefore, we tested whether *Tex19.1*, which is expressed in pluripotent cells (*Kuntz et al., 2008*), has a role in regulating L1 expression and restricting L1 mobilization in this cell type. We first investigated whether *Tex19.1* regulates mL1-ORF1p abundance in pluripotent mouse ESCs. Biochemical isolation of polyubiquitylated proteins suggests that endogenous mL1-ORF1p is polyubiquitylated in pluripotent mouse ESCs (*Figure 7A*). Furthermore, proteasome inhibition with lactacystin caused a ~4 fold increase in the abundance of mL1-ORF1p relative to $\beta$-actin after 6 hr of treatment (*Figure 7B*). Taken together these data suggest that mL1-ORF1p abundance is regulated by the proteasome in pluripotent mouse ESCs. hL1-ORF1p abundance is similarly regulated by the proteasome in human ESCs and human embryonal carcinoma (EC) cells (*Figure 7—figure supplement 1*). In contrast to a previous report assessing the abundance of retrotransposon RNA in ESCs derived from heterozygous mouse crosses (*Tarabay et al., 2013*), *Tex19.1*$^{-/-}$ mouse ESCs generated by sequential gene targeting (*Figure 7—figure supplement 2*) in a defined genetic background, cultured in 2i conditions, and analysed at low passage number do not de-repress L1 RNA (*Figure 7C*). These *Tex19.1*$^{-/-}$ mouse ESCs contain elevated levels of endogenous mL1-ORF1p, but this increase in mL1-ORF1p levels is not accompanied by increased endogenous L1 mRNA levels (*Figure 7C*). Moreover, loss of *Tex19.1* does not detectably affect transcription or translation of L1 reporter constructs in ESCs (*Figure 7—figure supplement 3*). Taken together these data suggest that, similar to male germ cells (*Figure 1*), *Tex19.1* functions to post-translationally repress mL1-ORF1p in pluripotent cells.

Next we tested whether loss of *Tex19.1* also results in increased mobilization of mouse L1 constructs in pluripotent ESCs. Although L1 retrotransposition assays have previously been performed in pluripotent human cells (*Wissing et al., 2011*; *Garcia-Perez et al., 2007*, *2010*), this assay has not yet been adapted to mouse ESCs and, to our knowledge, no restriction factor has been shown to restrict mobilization of L1 constructs in mouse pluripotent cells or germ cells. Therefore we optimized the L1 retrotransposition assay in mouse ESCs (García-Cañadas *et al.*, manuscript in preparation) using a neomycin retrotransposition indicator cassette (*Freeman et al., 1994*). Notably, the optimized assay routinely resulted in the appearance of hundreds of G418-resistant colonies when mouse ESCs were transfected with an active mouse $T_f$ L1 construct (*Han and Boeke, 2004*) (*Figure 7D*). Controls verified that co-transfection of the L1 restriction factor APOBEC3A (*Bogerd et al., 2006b*) severely reduces mL1 retrotransposition in mouse ESCs (*Figure 7—figure supplement 4A*), and that an allelic mL1 containing the N21A missense mutation in the EN domain of ORF2p (*Alisch et al., 2006*) retrotransposes at low levels in mouse ESCs (*Figure 7—figure supplement 4B*). Thus, the adapted L1 retrotransposition assay appears to reflect *bone fide* mobilization of L1 constructs in mouse ESCs. We next used the optimized assay to investigate the role of *Tex19.1* in controlling L1 retrotransposition in pluripotent mouse ESCs. Interestingly, mobilization of an active mouse $T_f$ L1 is reproducibly elevated around 1.5-fold in *Tex19.1*$^{-/-}$ ESCs relative to *Tex19.1*$^{+/+}$ wild-type ESCs (*Figure 7D*, *Figure 7—figure supplement 4B*). Control experiments revealed that both *Tex19.1*$^{+/+}$ and *Tex19.1*$^{-/-}$ ESCs could generate similar numbers of G418-resistant foci when transfected with a plasmid carrying a neomycin resistance cassette (*Figure 7—figure supplement 4C*). Thus, these data strongly suggest that *Tex19.1* controls L1 retrotransposition in mouse pluripotent ESCs, presumably at least in part by promoting proteasome degradation of mL1-ORF1p. To further test this, we analysed whether *Tex19.1* could restrict retrotransposition of an

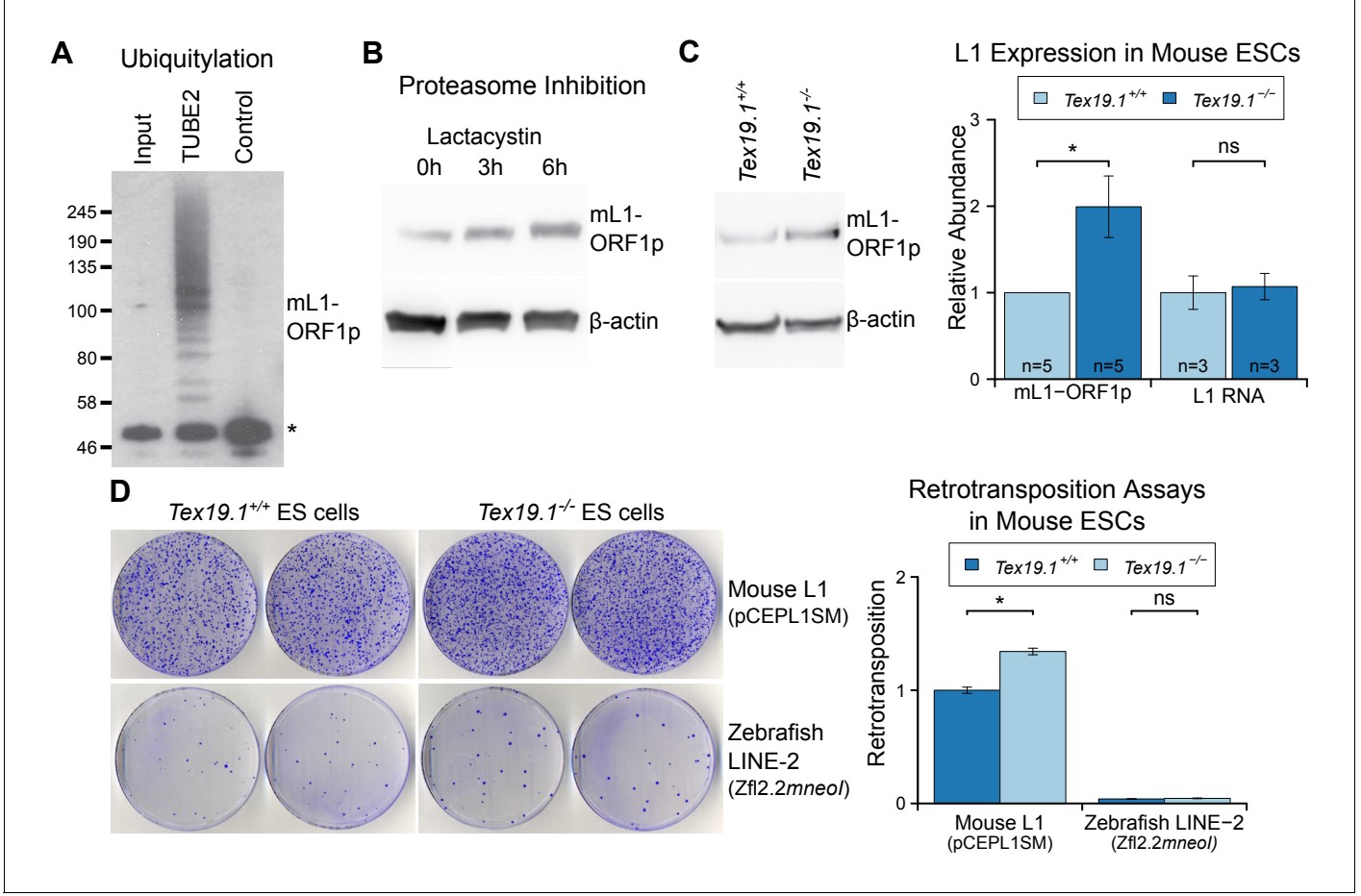

**Figure 7.** *Tex19.1* negatively regulates mL1-ORF1p abundance and L1 mobilization in mouse ESCs. (**A**) Mouse ESC lysates (input) were incubated with polyubiquitin-binding TUBE2 beads or control agarose beads and Western blotted for endogenous mL1-ORF1p. Non-specific binding of non-ubiquitylated mL1-ORF1p is detectable (asterisk), in addition to specific enrichment of polyubiquitylated mL1-ORF1p with TUBE2. (**B**) Western blot for endogenous mL1-ORF1p after treatment with 25 μM lactacystin proteasome inhibitor for the indicated times. $\beta$-actin is a loading control. (**C**) Western blot for endogenous mL1-ORF1p in *Tex19.1*$^{+/+}$ and *Tex19.1*$^{-/-}$ mouse ESCs. mL1-ORF1p abundance (Western blot) and L1 RNA abundance (qRT-PCR using primers against ORF2) were quantified relative to $\beta$-actin and normalized to *Tex19.1*$^{+/+}$ ESCs. Means ± SEM are indicated (1 ± 0 and 1.99 ± 0.36 for protein and 1 ± 0.19 and 1.07 ± 0.15 for RNA for *Tex19.1*$^{+/+}$ and *Tex19.1*$^{-/-}$ respectively); *p<0.05; ns indicates not significant (t-test, p=0.049, 0.8 from left to right). (**D**) Neomycin-resistant colonies from L1 retrotransposition assays in *Tex19.1*$^{+/+}$ and *Tex19.1*$^{-/-}$ ESCs. ESCs were transfected with LINE retrotransposition constructs carrying the *mneoI* indicator cassette and either synthetic mouse L1 (pCEPL1SM) or zebrafish LINE-2 (Zfl2.2) sequences, the number of neomycin-resistant colonies counted, and retrotransposition frequency calculated relative to *Tex19.1*$^{+/+}$ ESCs transfected with pCEPL1SM. *p<0.05; ns indicates not significant (t-test, p=0.01, 0.3 from left to right); error bars indicate SEM.

The following figure supplements are available for figure 7:

**Figure supplement 1.** hL1-ORF1p abundance in human embryonal carcinoma cells and human ESCs increases in response to inhibition of the proteasome.

**Figure supplement 2.** Generation and validation of *Tex19.1*$^{-/-}$ ESCs.

**Figure supplement 3.** Loss of *Tex19.1* does not affect L1 promoter or L1 translation reporter activity in mouse ESCs.

**Figure supplement 4.** *Tex19.1* restricts mobilization of engineered L1 constructs in mouse ESCs.

active zebrafish LINE-2 element that naturally lacks ORF1p but can efficiently retrotranspose in cultured human cells (*Sugano et al., 2006*; *Garcia-Perez et al., 2010*), and in cultured chicken cells that lack endogenous L1-ORF1p (*Suzuki et al., 2009*). Remarkably, loss of *Tex19.1* does not influence the rate of retrotransposition of the ORF1p-independent engineered zebrafish LINE-2 construct in mouse ESCs (*Figure 7D*). Thus, these data suggest that one role of endogenously expressed *Tex19.1* in mouse pluripotent cells is to restrict L1 mobilization, and thereby promote genome stability in the cells that can transmit new L1 integrations to the next generation.

## Discussion

This study identifies *Tex19.1* as a host restriction factor for L1 mobilization in the mammalian germline. We have previously reported that *Tex19.1* plays a role in regulating the abundance of retrotransposon RNAs (*Ollinger et al., 2008*; *Reichmann et al., 2012*, *2013*), which appears to reflect transcriptional de-repression of specific retrotransposons (*Crichton et al., 2017a*). Although loss of *Tex19.1* results in de-repression of L1 RNA in placenta (*Reichmann et al., 2013*), L1 RNA abundance is not affected by loss of *Tex19.1* in male germ cells (*Ollinger et al., 2008*) or, in contrast to a previous report (*Tarabay et al., 2013*), in mouse ESCs (*Figure 7*). Indeed here we show that *Tex19.1* has a role in the post-translational regulation of L1-ORF1p steady-state levels in these cells. Thus, *Tex19.1* appears to regulate retrotransposons at multiple stages of their life cycle. It is possible that *Tex19.1* is affecting different E3 ubiquitin ligases, or different E3 ubiquitin ligase substrates, in order to repress different stages of the retrotransposon life cycle. However, loss of *Tex19.1* results in a 1.5-fold increase in the rate of mobilization of L1 constructs in pluripotent cells. Since L1 mobilization mostly takes place in the pluripotent phase of the germline cycle, and new L1-dependent mobilization events are thought to be inherited by one in every twenty human births (*Kazazian, 1999*), *TEX19* activity could be having a significant impact on L1-derived mutations during evolution. Retrotransposons appear to provide functions that are advantageous for mammalian development and evolution (*Garcia-Perez et al., 2016*), and the activity of restriction mechanisms like the TEX19-dependent mechanism we have described here, that control the ability of retrotransposons to mobilize, rather than eliminate their transcriptional activity altogether, could potentially allow retrotransposons to participate in and drive the evolution of key gene regulatory networks in pluripotent cells while minimising their mutational load on the germline genome.

Our data suggests that L1-ORF1p is post-translationally modified by ubiquitylation in somatic and germline cells. Phosphorylation of L1-ORF1p has been previously reported in somatic tissues and is required for L1 retrotransposition in these cells (*Cook et al., 2015*). However, we are not aware of any previous reports that post-translational modifications of L1-ORF1p are present in the germline, particularly in the pluripotent phase of the germline cycle when L1 retrotransposition is thought to primarily occur (*Kano et al., 2009*). There are 32 lysine residues in human L1-ORF1p that could act as potential ubiquitylation sites (*deHaro et al., 2014*), and 42, 47 and 39 lysines in mouse $T_f$, $G_f$ and A subtypes of L1-ORF1p respectively that could act as potential ubiquitylation sites. It will be of interest to determine which of these lysines are ubiquitylated in somatic and germline tissues, and how variant these residues are between retrotransposition-competent L1s. Post-translational regulation of L1 potentially provides an additional layer of genome defence that could be particularly important during periods of epigenetic reprogramming in early embryogenesis or in the developing primordial germ cells when transcriptional repression of retrotransposons might be more relaxed (*Molaro et al., 2014*; *Fadloun et al., 2013*). Indeed, the sensitivity of *Tex19.1* expression to DNA hypomethylation (*Hackett et al., 2012*) will allow post-translational suppression of L1 to be enhanced during these stages of development. Post-translational regulation of L1s is also likely important to limit the activity of L1 variants that evolve to escape transcriptional repression by the host and will provide a layer of genome defence while the host adapts its KRAB zinc-finger protein repertoire to these new variants (*Jacobs et al., 2014*). Analysis of L1 evolution shows that regions within L1-ORF1p are under strong positive selection suggesting that host restriction systems are targeting L1-ORF1p post-translationally and impacting on evolution of these elements (*Boissinot and Furano, 2001*; *Sookdeo et al., 2013*). Although this evidence for post-translational restriction factors acting on L1-ORF1p has been known for over 15 years, to our knowledge no host factors have been identified that directly bind to L1-ORF1p and restrict L1 mobilization in germline cells. It is possible that the physical interactions between L1-ORF1p and TEX19:UBR2 that we describe here are

contributing to these selection pressures acting on L1-ORF1p. While UBR2 is able to target L1-ORF1p in the absence of TEX19, evolution of a less constrained TEX19 adapter to provide a further link between UBR2 and L1-ORF1p could potentially resolve the contradictory pressures on UBR2 to maintain interactions with some endogenous cellular substrates while targeting a rapidly evolving retrotransposon protein for degradation.

Our data strongly suggest that TEX19.1 likely exists in a complex with UBR2 in ESCs, and that TEX19.1 stimulates a basal activity of UBR2 to promote polyubiquitylation of L1-ORF1p (*Figure 8*). Ubr1, a yeast ortholog of UBR2, has different binding sites for different types of substrate (*Xia et al., 2008*). Ubr1 participates in the N-end rule pathway that degrades proteins depending on their N-terminal amino acids, and can bind to and ubiquitylate proteins containing specific residues at their N-termini (N-end rule degrons). Ubr1 also binds to and catalyses ubiquitylation of proteins that have more poorly defined non-N-terminal internal degrons (*Xia et al., 2008*; *Sriram et al., 2011*; *Kim et al., 2014*). Full-length human L1-ORF1p does not have a potential N-end rule degron at its N-terminus (*Kim et al., 2014*; *Sriram et al., 2011*), and we speculate the interaction between UBR2 and L1-ORF1p likely reflects an internal degron in the retrotransposon protein. One of the known internal degron substrates of yeast Ubr1 is CUP9, a transcription factor that regulates expression of a peptide transporter (*Turner et al., 2000*). Binding and polyubiquitylation of CUP9 by Ubr1 is allosterically activated by specific dipeptides binding to the N-end rule degron binding sites in Ubr1 (*Du et al., 2002*; *Xia et al., 2008*; *Turner et al., 2000*). The effect of these dipeptides on Ubr1 activity in yeast strongly resonates with the effects of TEX19 orthologs on UBR2 activity in mammals: TEX19 orthologs binds to UBR2 and inhibits its activity towards N-end rule substrates (*Reichmann et al., 2017*), but stimulate polyubiquitylation of L1-ORF1p. The direct interaction between TEX19 orthologs and L1-ORF1p could further enhance L1-ORF1p binding to UBR2 by stabilizing the highly flexible L1-ORF1p trimers (*Khazina et al., 2011*) in a conformational state that exposes an internal degron and favors their ubiquitylation. Thus, TEX19 orthologs appear to function, at least in part, by re-targeting UBR2 away from N-end rule substrates and towards a

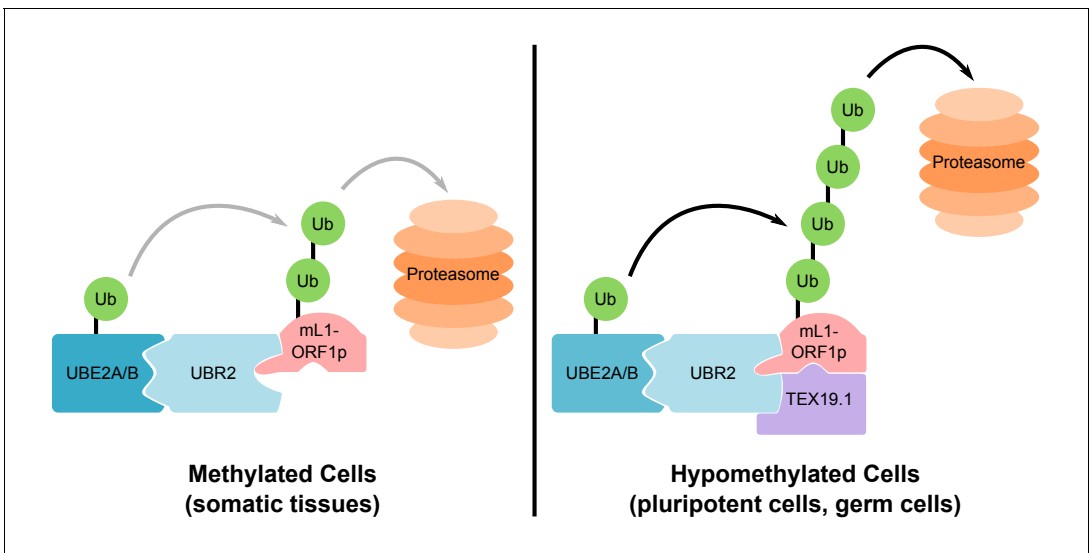

**Figure 8.** Model For UBR2 and TEX19.1-mediated polyubiquitylation of mL1-ORF1p. In methylated somatic cells, the RING domain E3 ubiquitin ligase UBR2 and its cognate E2 ubiquitin conjugating enzyme UBE2A/B can interact with mL1-ORF1p and catalyse ubiquitylation and proteasome-dependent turnover of this protein. TEX19.1 in hypomethylated cells, including pluripotent cells and germ cells, interacts with both UBR2 and mL1-ORF1p, stimulating further polyubiquitylation and proteasome-dependent turnover of mL1-ORF1p. The interaction between TEX19.1 and UBR2 concomitantly inhibits the activity of UBR2 towards N-end rule substrates (*Reichmann et al., 2017*). This model does not exclude additional factors and/or mechanisms contributing to the effects of UBR2 and TEX19.1 on the stability of mL1-ORF1p.

The following figure supplement is available for figure 8:

**Figure supplement 1.** Model for retrotransposon regulation during epigenetic reprogramming in lineage-restricted germ cells.

retrotransposon substrate. However, the direct interaction between TEX19 orthologs and L1-ORF1p means that it is possible that TEX19 orthologs are interfering with L1-ORF1p function in multiple ways in order to restrict L1 mobilization. Thus, while one outcome of this interaction appears to be increased polyubiquitylation and degradation of L1-ORF1p, the interaction between TEX19 orthologs and L1-ORF1p could also interfere with the nucleic acid chaperone activity of L1-ORF1p (*Martin et al., 2005*), or its interactions with either L1-encoded or host-encoded molecules (*Taylor et al., 2013*; *Goodier et al., 2013*).

The constellation of L1 sequences in the genome (*Chinwalla et al. 2002*) makes it difficult to quantitatively determine how much each L1 locus contributes to the cellular pool of L1 RNAs, and how much each L1 RNA contributes to the amount of L1-encoded proteins in the cell. We have been unable to detect effects on bulk transcription of L1 in the absence of *Tex19.1*, and *Tex19.1* could be potentially regulating endogenous L1-ORF1p abundance in testes and ES cells entirely post-transcriptionally. However, we cannot rule out the possibility that transcriptional or translational derepression of specific variant copies of L1 are contributing to the increase in the abundance of L1-ORF1p species detected in $Tex19.1^{-/-}$ ES cells and testes. Our data using tagged copies of L1-ORF1p have allowed us to link transcription and protein abundance from a single defined L1 sequence suggesting that *Tex19.1* can act, at least in part, at post-transcriptional level to regulate endogenous L1-ORF1p abundance in the germline.

Our data are consistent with TEX19.1 playing a role in promoting polyubiquitylation of mL1-ORF1p in mouse ESCs, thereby reducing the steady-state abundance of mL1-ORF1p in these cells (*Figure 7*). Quantifying the amount of the transient heterogeneous mixture of polyubiquitylated mL1-ORF1p endogenously expressed in control and $Tex19.1^{-/-}$ mESCs cells is technically challenging. This is partly due to the activity of deubiquitylases present in the ES cell lysates, partly due to the heterogeneous nature of endogenously expressed mL1-ORF1p, which may be recognised with multiple different affinities by anti-mL1-ORF1p antibodies, particularly when present in different ubiquitylated states, and partly because this experiment would likely require ESCs to be treated with proteasome inhibitor to allow polyubiquitylated species to accumulate. This treatment can stabilize E3 ubiquitin ligases like UBR2 itself (*An et al., 2012*), or other proteins that can regulate L1-ORF1p abundance independently of TEX19.1. Cell-based ubiquitylation assays (*Figure 4*) circumvent these challenges by assessing the effect of TEX19 on a single epitope-tagged copy of L1-ORF1p in the absence of intervention with proteasome inhibitors and under denaturing conditions that inactivate deubiquitylases in the lysate. Taken together, the protein interactions, gain-of-function cell-based ubiquitylation data, and loss-of-function phenotyping in ESCs and in mouse testes indicate that TEX19.1 plays a role in regulating the polyubiquitylation and stability of mL1-ORF1p.

The data presented here suggests that programmed DNA hypomethylation in the mouse germline extends beyond activating components of the PIWI-piRNA pathway (*Hackett et al., 2012*) to include enhancing the activity of the ubiquitin-proteasome system towards retrotransposon substrates. Recent data has suggested that TEX19.1 physically interacts with components of the PIWI-piRNA pathway (*Tarabay et al., 2017*), although it is not clear whether these proposed interactions have functional consequences for retrotransposon suppression *in vivo*. Activation of post-translational genome-defence mechanisms may allow mammalian germ cells to safely transcribe retrotransposons by preventing these transcripts from generating RNPs that can mutate the germline genome (*Figure 8—figure supplement 1*). The retrotransposon transcripts can then potentially be processed into piRNAs and used to identify retrotransposon loci where epigenetic silencing needs to be established. *De novo* establishment of epigenetic silencing at retrotransposons in the *Arabidopsis* germline involves transfer of small RNAs between a hypomethylated vegetative cell and a germ cell (*Slotkin et al., 2009*), whereas these processes happen sequentially in the same germ cell in mammals (*Figure 8—figure supplement 1*). Therefore the ability to enhance post-translational control of retrotransposons may be a key feature of epigenetic reprogramming in the mammalian germline that limits the trans-generational genomic instability caused by retrotransposon mobilization.

## Materials and methods

### Mice

*Tex19.1* mutant mice (RRID:MGI:4453205) on a C57BL/6J genetic background (RRID:IMSR_JAX: 000664, obtained from Charles River) were maintained and genotyped as described (*Ollinger et al., 2008*; *Reichmann et al., 2012*). *Tex19.1*[+/−] heterozygotes have no detectable testis phenotype and indistinguishable sperm counts from wild-type animals (*Ollinger et al., 2008*), and prepubertal *Tex19.1*[−/−] homozygotes were typically compared with heterozygous littermates to control for variation between litters. *Ubr2*[−/−] mice were generated by CRISPR/Cas9 double nickase-mediated genome editing in zygotes (*Ran et al., 2013*). Complementary oligonucleotides (*Supplementary file 3*) targeting exon 3 of UBR2 were annealed and cloned into plasmid pX335 (*Cong et al., 2013*), amplified by PCR, then in vitro transcribed using a T7 Quick High Yield RNA Synthesis kit (NEB) to generate paired guide RNAs. RNA encoding the Cas9 nickase mutant (50 ng/μl, Tebu-Bio), paired guide RNAs targeting exon 3 of UBR2 (each at 25 ng/μl), and 150 ng/μl single-stranded DNA oligonucleotide repair template (*Supplementary file 3*) were microinjected into the cytoplasm of B6CBAF1/J × B6CBAF1/J zygotes (RRID:IMSR_JAX:100011, obtained from Charles River). The repair template introduces an *Xba*I restriction site and mutates cysteine-121 within the UBR domain of UBR2 (Uniprot Q6WKZ8-1) to a premature stop codon. The zygotes were then cultured overnight in KSOM (Millipore) and transferred into the oviduct of pseudopregnant recipient females. Pups were genotyped for the presence of the *Xba*I restriction site. The *Ubr2*[−/−] male mice generated in this way have no overt phenotypes except testis defects and infertility and *Ubr2*[−/−] females are born at sub-Mendelian ratios, all as previously described for *Ubr2*[−/−] mice generated by gene targeting in ESCs (*Kwon et al., 2003*). The day of birth was designated P1, and mice were culled by cervical dislocation. Mouse experiments were performed in accordance with local ethical guidelines and under authority of UK Home Office Project Licence PPL 60/4424. For mouse experiments, a sample size of three mutant animals was typically used and alongside littermate controls to allow consistent phenotypic changes in retrotransposon expression to be associated with genotype. Each animal was considered a biological replicate.

### Cell culture

We used cell lines that were previously shown to support retrotransposition of engineered L1 constructs or *Tex19.1*[−/−] models generated in this study. Cell lines were maintained at 37°C in 5% CO$_2$. HEK293T and U2OS cells were obtained from ATCC (ATCC Cat# CRL-3216, RRID:CVCL_0063; ATCC Cat# HTB-96, RRID:CVCL_0042) and HeLa cells were provided by John V. Moran (University of Michigan, US). These cell lines were grown in Dulbecco's Modified Eagle's Media (DMEM) supplemented with 10% foetal calf serum, 1% penicillin-streptomycin, and 1% L-glutamine. E14Tg2a mouse ESCs (RRID:CVCL_9108) were obtained from Julia Dorin (MRC Human Genetics Unit, UK) and cultured on gelatinized flasks in 2i culture conditions (1:1 DMEM/F12 media:neurobasal media supplemented with N2 and B27, 10% foetal calf serum, 1% L-glutamine, 0.1% *β*-mercaptoethanol, 1 μM PD0325901 (StemMACS), and 3 μM CHIR99021 (StemMACS).

Hamster XR-1 cells (RRID:CVCL_K253) (*Stamato et al., 1983*) were provided by Thomas D. Stamato (The Lankenau Institute fro Medical Research, US) and grown in DMEM low glucose medium containing 10% foetal calf serum, 1% L-glutamine, 1% penicillin-streptomycin and 0.1 mM nonessential amino acids. Human PA-1 cells (*Zeuthen et al., 1980*) were obtained from ATCC (ATCC Cat# CRL-1572, RRID:CVCL_0479) and grown in Minimal Essential Media (MEM) supplemented with 10% heat-inactivated foetal calf serum, 1% L-glutamine, 1% penicillin-streptomycin and 0.1 mM nonessential amino acids. H9 human ESCs (*Thomson et al., 1998*) were obtained from Wicell (RRID: CVCL_9773) and cultured and passaged as previously described using conditional media (CM) (*Garcia-Perez et al., 2007*). To prepare CM, human foreskin fibroblasts obtained from ATCC (ATCC Cat# SCRC-1041, RRID:CVCL_3285) were mitotically inactivated with 3000–3200 rads γ-irradiation, seeded at 3 × 10$^6$ cells/225 cm$^2$ flask and cultured with hESC media (KnockOut DMEM supplemented with 4 ng/ml bFGF, 20% KnockOut serum replacement, 1 mM L-glutamine, 0.1 mM *β*-mercaptoethanol and 0.1 mM non-essential amino acids) for at least 24 hr before media harvesting. We collected CM 24, 48 and 72 hr after seeding. H9 human ESCs (Wicell, RRID:CVCL_9773) were maintained on Matrigel (BD Biosciences)-coated plates in human foreskin fibroblast-conditioned media.

The absence of *Mycoplasma* in cultured cells was confirmed once a month using a PCR-based assay (Minerva). Single tandem repeat genotyping was done at least once a year to ensure the identity of the human cell lines used. The identity of parental mouse ESCs was confirmed by generation of chimeric mice and germline transmission, and parental and targeted mouse ESCs were confirmed to contain forty chromosomes by karyotyping. The identity of hamster XR-1 cells was confirmed using an endonuclease-independent retrotransposition assay (*Morrish et al., 2002*). Independent wells, plates or transfections were used as biological replicates.

## Generation of stable cell lines

ESCs and HEK293 cell lines stably expressing TEX19.1-YFP or YFP alone were generated by transfecting E14Tg2a ESCs or HEK293 cells with linearized pCAG-TEX19.1-YFP and pCAG-YFP expression plasmids (*Supplementary file 4*) containing the CAG promoter for expression (*Niwa et al., 1991*), and selecting for the G418 resistance cassette. Stable cell lines were flow sorted to select for YFP expression. For pCAG-YFP transfection, the cell lines were flow sorted to select for cells expressing YFP at similar levels to the pCAG-TEX19.1-YFP cell lines. Stable Flp-In-293 cells (Invitrogen) expressing T7-tagged hL1-ORF1p from a CMV promoter at the Flp-In locus were generated using the pcDNA5/FRT Flp-In vector, and selected using 100 µg/ml hygromycin and 100 µg/ml Zeocin according to the supplier's instructions.

## Generation of *Tex19.1*$^{-/-}$ ESCs

*Tex19.1*$^{-/-}$ ESCs were generated by sequential targeting of E14Tg2a ESCs. The *Tex19.1* targeting vector was generated by inserting an IRES-GFP cassette into position chr11:121147942 (mm10 genome assembly) in the 3′ untranslated region of *Tex19.1* in a bacterial artificial chromosome (BAC) by BAC recombineering (*Liu et al., 2003*). A 13 kb region (chr11:121143511–121156687) containing *Tex19.1* was gap-repaired into PL253 (*Liu et al., 2003*), then a LoxP site from PL452 was recombined upstream of the coding exon at position chr11:121146376, and an Frt-flanked neomycin-resistance cassette and second LoxP site from PL451 (*Liu et al., 2003*) recombined downstream of the coding exon at chr11:121148877. E14Tg2a ESCs were electroporated with the resulting targeting vector, selected for neomycin resistance, and correct integrants identified by PCR. The *Tex19.1* coding exon in the targeted allele was removed by transfection with a Cre-expressing plasmid, and the resulting cells electroporated with the targeting vector again, selected for neomycin resistance, and correct integrants on the second *Tex19.1* allele identified by PCR. ESCs were then transiently transfected with a Flp-expressing plasmid to generate a conditional *Tex19.1*$^{fl}$ allele. This was subsequently converted to a *Tex19.1*$^-$ allele by transient transfection with a Cre-expressing plasmid to remove the *Tex19.1* coding exon. ESCs were cultured in gelatinized flasks in LIF+serum (Glasgow Modified Eagle's Media, 10% foetal calf serum, 1% non-essential amino acid, 1% sodium pyruvate, 1% penicillin-streptomycin, 1% L-glutamine, 0.001% $\beta$-mercaptoethanol, and 0.2% leukaemia inhibitory factor-conditioned media) during the generation of *Tex19.1*$^{-/-}$ ESCs, then low passage *Tex19.1*$^{-/-}$ ESCs with a euploid karyotype were used for experiments after transitioning to 2i culture conditions for at least 14 days.

## qRT-PCR

RNA was isolated from cells or tissues using TRIzol reagent (Life Technologies), treated with DNAse (DNAfree, Ambion) and used to generate random-primed cDNA (First Strand cDNA Kit, Life Technologies) as described by the suppliers. qPCR was performed on the cDNA using the SYBR Green PCR System (Stratagene) and a CFX96 Real-Time PCR Detection System (Bio-Rad). Control qRT-PCR reactions were performed in the absence of either reverse transcriptase or qPCR template to verify the specificity of any qRT-PCR signals obtained. Primers were validated to perform at >90% efficiency in the qRT-PCR assay, and expression quantified using the $2^{-\Delta\Delta Ct}$ method (*Livak and Schmittgen, 2001*). Alternatively, qPCR was performed using SYBR Select Master Mix (Applied Biosystems) and a Light Cycler 480 II (Roche), and expression quantified using the relative standard curve method as described by the suppliers. Sequences of oligonucleotide primers used for qRT-PCR are listed in *Supplementary file 3*.

## Western blotting

Tissue or cells were homogenized in 2× Laemmli SDS sample buffer (Sigma) with a motorized pestle, then boiled for 2–5 min and insoluble material pelleted in a microcentrifuge. Protein samples were resolved on pre-cast Bis-Tris polyacrylamide gels in MOPS running buffer (Invitrogen), or Tris-Acetate polyacrylamide gels in Tris-Acetate SDS running buffer (Invitrogen) and Western blotted to PVDF membrane using a GENIE blotter (Idea Scientific) or the iBlot Transfer system (Invitrogen). Pre-stained molecular weight markers (Thermo Fisher) were used to monitor electrophoresis and blotting. Membranes were blocked with 5% non-fat skimmed milk powder in PBST (PBS, 0.1% Tween-20), then incubated with primary antibodies (*Supplementary file 5*) diluted in blocking solution. Membranes were then washed with PBST and, if required, incubated with peroxidase-conjugated secondary antibody in blocking solution. Membranes were washed in PBST and bound secondary antibodies detected using West Pico Chemiluminescent Substrate (Thermo Scientific). Western blots were quantified using ImageJ (*Schneider et al., 2012*). For simultaneous two-color detection and quantification, proteins were transferred to nitrocellulose membranes, rabbit L1-ORF1p antibodies were used at a 1:1000 dilution and mouse $\beta$-actin at 1:2000, and IRDye-conjugated secondary antibodies (LI-COR) detected using an Odyssey imager (LI-COR).

## Immunostaining

Immunostaining on P16 testes was performed by fixing decapsulated P16 testes in 4% paraformaldehyde in PBS, embedding the tissue in paraffin wax, and cutting 6 µm sections on a microtome. Sections were de-waxed in xylene, rehydrated, and antigen retrieval was performed by boiling slides in a microwave for 15 mins in 10 mM sodium citrate pH 6. Sections were blocked in PBS containing 10% goat serum, 3% BSA, 0.1% Tween-20, then incubated in 1:300 rabbit anti-mL1-ORF1p primary antibody (*Martin and Branciforte, 1993*; *Soper et al., 2008*) diluted in blocking solution. Sections were then washed with PBS, incubated in 1:500 Alexa Fluor-conjugated secondary antibodies (Life Technologies), washed with PBS again, then mounted under a coverslip using Vectashield mounting media containing DAPI (Vector Laboratories). Slides were imaged on a Zeiss Axioplan II fluorescence microscope equipped with a Hamamatsu Orca CCD camera. Anti-mL1-ORF1p fluorescence intensity was measured per unit area, with slides immunostained with non-specific rabbit IgG and secondary antibodies used to calculate and subtract background.

## Polysome gradients

Polysome gradients were prepared as described (*Gillian-Daniel et al., 1998*). In brief, P18 testes were homogenized in 200 µl lysis buffer (20 mM HEPES pH 7.4, 150 mM KCl, 5 mM DTT, 5 mM MgCl$_2$, 100 U/mL RNasein, Complete protease inhibitors (Roche), 10 nM calyculin A, 150 µg/mL cycloheximide) then NP-40 added to 0.5% and the samples incubated on ice for 10 min. After centrifugation at 12,000 g for 5 min at 4°C the soluble supernatant was layered onto an 11 mL 10–50% linear sucrose prepared in gradient buffer (20 mM HEPES pH 7.4, 250 mM KCl, 5 mM DTT, 10 mM MgCl$_2$, 1 µg/µL heparin), then centrifuged in a SW41Ti rotor (Beckman) for 120 min at 38,000 rpm at 4°C. 1 mL fractions were collected and absorbance of RNA at 254 nm was recorded by using a UV monitor. To isolate RNA, fractions were digested with 20 µg/µL proteinase K in presence of 1% SDS and 10 mM EDTA for 30 min at 37°C then RNAs recovered using Trizol LS reagent (Invitrogen). To isolate proteins, fractions were precipitated with methanol/chloroform and pellets resuspended by boiling in Laemmli SDS sample buffer.

## Oligo(dT) pull-downs

P16 testes were homogenized with a motorized pestle in lysis buffer (20 mM HEPES pH 7.4, 150 mM KCl, 5 mM DTT, 5 mM MgCl$_2$) supplemented with 100 U/mL RNasein, Complete protease inhibitors (Roche) and insoluble debris removed by centrifugation (12,000 g, 5 min at 4°C). Oligo(dT)-cellulose beads (Ambion) were blocked in lysis buffer containing 5% BSA for 1 hr at 4°C, then incubated with lysate for 1 hr at 4°C. Oligo(dT)-cellulose beads were washed three times with lysis buffer, and bound proteins eluted by boiling in Laemmli SDS sample buffer and analysed by Western blotting. For competition assays, 200 µg of a 25-mer poly(A) oligonucleotide (Sigma Genosys) was incubated with the oligo(dT)-cellulose beads for 30 min at 4°C before the addition of lysates. Poly(A) binding protein PABP1 was used as a positive control (*Burgess et al., 2011*).

## Isolation of TEX19.1-YFP complexes

Cytoplasmic extracts were prepared as described (*Wright et al., 2006*). Briefly, stable YFP or TEX19.1-YFP ESCs growin in LIF+serum conditions were resuspended in three volumes buffer A (10 mM HEPES pH 7.6, 15 mM KCl, 2 mM MgCl$_2$, 0.1 mM EDTA, 1 mM DTT, 0.2 mM PMSF, Complete protease inhibitors (Roche)) and incubated on ice for 30 mins. Cells were lysed in a Dounce homogenizer, one-tenth volume buffer B (50 mM HEPES pH 7.6, 1 M KCl, 30 mM MgCl$_2$, 0.1 mM EDTA, 1% NP-40, 1 mM DTT, 0.2 mM PMSF) added, then the lysate centrifuged twice for 15 min at 3400 g at 4°C to deplete nuclei. Glycerol was added to a final volume of 10%, the extracts centrifuged at 12,000 g for 5 min at 4°C, pre-cleared with protein A agarose beads (Sigma) then with blocked agarose beads (Chromotek), before incubation with GFP-Trap agarose beads (ChromoTek Cat# gta-20 RRID:AB_2631357) for 90 min at 4°C. Beads were collected by centrifugation at 2700 g for 2 min at 4°C, washed three times with 9:1 buffer A:buffer B, and protein eluted by boiling in 2× Laemmli SDS sample buffer for 3 min. Protein samples were separated on pre-cast Bis-Tris polyacrylamide gels (Invitrogen) and stained with Novex colloidal blue staining kit (Invitrogen). Lanes were cut into seven regions according to migration of molecular weight markers and in-gel digestion with trypsin, and mass spectrometry using a 4800 MALDI TOF/TOF Analyser (ABSciex) equipped with a Nd:YAG 355 nm laser was performed by St. Andrews University Mass Spectrometry and Proteomics Facility. Mass spectrometry data was analysed using the Mascot search engine (Matrix Science) to interrogate the NCBInr database using tolerances of ± 0.2 Da for peptide and fragment masses, allowing for one missed trypsin cleavage, fixed cysteine carbamidomethylation and variable methionine oxidation.

## Size exclusion chromatography

Superdex 200 10/300 GL (GE Healthcare Life Sciences) was calibrated with molecular weight markers for gel filtration (Sigma-Aldrich) in BC200 buffer (25 mM HEPES pH 7.3, 200 mM NaCl, 1 mM MgCl$_2$, 0.5 mM EGTA, 0.1 mM EDTA, 10% glycerol, 1 mM DTT, and 0.2 mM PMSF). 2 mg cytoplasmic extract from ESCs grown in LIF+serum were diluted in 500 µl buffer A/B (15 mM HEPES pH7.6, 115 mM KCl, 3 mM MgCl$_2$, 0.1 mM EDTA, 1 mM DTT, 0.2 mM PMSF, Complete protease inhibitors (Roche)) containing 20 µg RNase Inhibitor (Promega), centrifuged (12,000 g, 10 min at 4°C), then loaded on the column. The column was run isocratically in BC200 buffer for 1.4 column volumes and 0.5 ml fractions were collected. Fractions were precipitated with trichloroacetic acid and resuspended in Laemmli SDS sample buffer. Data shown is representative of two replicates.

## Co-Immunoprecipitation

HEK293T cells were transfected with plasmids (pCAG-*Tex19.1*-YFP, pCAG-*TEX19*-YFP, pEGFP3N1-*Ubr2*, pCMV5-hORF1-T7, pCMV5-mORF1-T7, pCMV5-mORF1-mCherry, pCMV5-hORF1$^{RA}$-T7, pCMV5-mORF1$^{RA}$-T7, *Supplementary file 4*) using Lipofectamine 2000 (Invitrogen) according to the manufacturer's instructions and incubated for 24 hr before harvesting. GFP-Trap agarose beads (Chromotek) were used to immunoprecipitate YFP- or GFP-tagged proteins following manufacturer's instructions. RFP-Trap agarose beads (ChromoTek Cat# rta-20 RRID:AB_2631362) was similarly used to immunoprecipitate mCherry-tagged proteins (*Shaner et al., 2005*), with the addition of a pre-clearing step using binding control agarose beads (Chromotek). The ORF1$^{RA}$ mutants contain two mutations in the RNA binding domain of L1-ORF1p (R260A and R261A in hL1-ORF1p, R297A and R298A in mL1-ORF1p) that reduce the ability of L1-ORF1p to bind RNA and form a RNP (*Kulpa and Moran, 2005*; *Martin et al., 2005*). These mutations abolish the ability of engineered L1 constructs to retrotranspose (*Figure 2—figure supplement 1F*) (*Moran et al., 1996*).

For anti-FLAG immunoprecipitation, cell pellets were lysed for 20 min on ice in lysis buffer (10 mM Tris pH 7.5, 150 mM NaCl, 0.5 mM EDTA, 0.5% NP-40, 1 mM PMSF, Complete Protease Inhibitors (Roche)), and insoluble material removed by centrifugation at 12,000 g for 10 min at 4°C. Supernatants were diluted 1:4 in lysis buffer without NP-40, then combined with washed anti-FLAG M2 affinity gel (Sigma-Aldrich Cat# A2220 RRID:AB_10063035), and rotated at 4°C for 1 hr. The anti-FLAG gel was washed three times in lysis buffer without NP-40, then protein eluted in 2× Laemmli SDS sample buffer for Western blot analysis. For all co-immunoprecipitation data, data shown is representative of at least two replicates.

## Cell-based ubiquitylation assays

HEK293T cells were cotransfected with equal amounts of the indicated plasmids (pCMV-*TEX19*, pCMV-His$_6$-myc-ubiquitin (*Ward et al., 1995*), and pCMV5-hORF1-T7, *Supplementary file 4*) using Lipofectamine 2000 (Invitrogen). Cells were harvested 72 hr after transfection and lysed in 6 M guanidinium-HCl, 0.1 M Na$_2$HPO$_4$, 0.1 M NaH$_2$PO$_4$, 0.01 M Tris-HCl pH 8.0, 5 mM imidazole and 10 mM $\beta$-mercaptoethanol. Following sonication, samples were rotated with washed Ni-NTA agarose (Qiagen) at room temperature for 4 hr. The agarose beads were washed as described (*Rodriguez et al., 1999*) and ubiquitylated proteins eluted with 200 mM imidazole, 0.15 M Tris-HCl pH 6.7, 30% glycerol, 0.72 M $\beta$-mercaptoethanol and 5% SDS then analysed by Western blotting. Data shown is representative of three replicates.

## TUBE2 pull-downs

E14Tg2a ES cells were lysed (50 mM Tris pH 7.5, 0.15 M NaCl, 1 mM EDTA, 1% NP-40, 10% glycerol, 5 mM N-ethylmaleimide, Complete Protease Inhibitors (Roche)) on ice for 20 min. Cell lysates were centrifuged at 12,000 g for 10 min at 4°C and soluble supernatant incubated at 4°C overnight with TUBE2 or control agarose (LifeSensors) prepared according to manufacturer's instructions. Agarose beads were washed three times in 50 mM Tris pH 7.4, 150 mM NaCl, 0.1% Tween and protein eluted with 2× Laemmli SDS sample buffer. Data shown is representative of three replicates.

## Strep pull-down from bacterial lysates

For the Strep pull-down assays, hL1-ORF1p and human TEX19 were either co-expressed or separately expressed overnight at 20°C in *Escherichia coli* BL21 (DE3) Star cells. Expression plasmids (*Diebold et al., 2011*), including a GB1 solubility tag (*Cheng and Patel, 2004*) for TEX19, are described in *Supplementary file 4*. The cells were lysed in a binding buffer (50 mM Hepes pH 7.0, 200 mM NaCl, 2 mM DTT) containing DNase I, lysozyme and protease inhibitors. For proteins expressed separately, 200 µl of the Strep tagged binding partner (hL1-ORF1p or GB1) were incubated with 50 µl Strep-Tactin Sepharose beads (IBA) in a total volume of 1 ml of binding buffer for 45 min at 4°C. After centrifugation (~1500 g) and two washes with 700 µl of binding buffer, 1 ml of TEX19 lysate was added to the beads, followed by an additional incubation for 45 min at 4°C. For co-expressed proteins, 1 ml of the lysate was added to 50 µl Strep-Tactin Sepharose beads (IBA) and incubated for 45 min at 4°C. In the end, the beads were washed five times with 700 µl of binding buffer. The bound proteins were eluted with 100 µl of the binding buffer supplemented with 2.5 mM biotin. The eluted proteins were then precipitated by trichloroacetic acid, resuspended in 1x SDS-PAGE sample buffer and analyzed by SDS-PAGE. For pull-downs with fragments of hL1-ORF1p and TEX19, proteins were always co-expressed as described above. Gel loading volumes were adjusted to obtain approximately equal amounts of bait protein in each lane.

For the treatment with micrococcal nuclease, co-expressed hL1-ORF1p and TEX19 were lysed in binding buffer (50 mM Hepes pH 7.0, 150 mM NaCl, 2 mM DTT) containing DNase I, lysozyme and protease inhibitors. After centrifugation for 30 min at 14000 g at 4°C, CaCl$_2$ was added to the final concentration of 2.5 mM to the lysate. To one half of the lysate micrococcal nuclease was added to the final concentration of $4 \times 10^3$ gel units/ml. The lysate was incubated for 15 min at 4°C, then centrifuged for 15 min at 18000 g at 4°C. The supernatant was then added to Strep-Tactin beads (IBA) as described above. Under these conditions, 4000 gel units/ml MNase entirely degrades 50 ug/ml oligo(A)$_{27}$ RNA.

## Luciferase assays

Luciferase activity was measured 24 hr post-transfection using the Dual-Luciferase Reporter Assay system (Promega) following manufacturer's instructions and as described previously (*Heras et al., 2013*).

## Retrotransposition assays

We used three different L1 retrotransposition assays in HEK293T, U2OS, HeLa and mouse ESCs. In all retrotransposition assays, we confirmed that overexpression of human *TEX19* or mouse *Tex19.1* is not toxic to cultured HeLa, HEK293T or U2OS cells. Where indicated, transfection efficiency controls were used to calculate rates of engineered retrotransposition as described (*Garcia-Perez et al.,*

*2010*; *Kopera et al., 2016*), and engineered L1 retrotransposons were co-transfected with a second expression plasmid for *TEX19* orthologs or controls. For *mneoI* and *mblastI*-based assays, we included a plasmid containing a neomycin or blasticidin resistance expression cassette respectively, to control for cytotoxicity (*Kopera et al., 2016*; *Richardson et al., 2014b*) when over-expressing *TEX19* orthologs.

Retrotransposition assays with *mneoI* or *mblastI* tagged L1 constructs in cultured HeLa and U2OS cells were performed as described (*Kopera et al., 2016*; *Morrish et al., 2002*; *Richardson et al., 2014b*; *Wei et al., 2000*). L1 constructs used in these assays were derived from active human L1 elements (*Brouha et al., 2002*; *Kimberland et al., 1999*; *Sassaman et al., 1997*; *Moran et al., 1996*), active mouse L1 elements (*Goodier et al., 2001*), or a synthetic codon-optimized mouse L1 element (*Han and Boeke, 2004*), and are described in *Supplementary file 4*. HeLa cells were transfected with Fugene6 (Promega) using 1 µg plasmid DNA per 35 mm diameter well and OptiMEM (Invitrogen) according to the manufacturer instructions. 400 µg/ml G418 selection for 12 days was initiated 72 hr post-transfection for *mneoI* constructs, or 10 µg/ml blasticidin S selection was initiated 120 hr post-transfection for 7 days for *mblastI* constructs. Drug-resistant foci were then fixed (2% formaldehyde, 0.2% glutaraldehyde in PBS) and stained (0.1% crystal violet). Retrotransposition assays with *mneoI* tagged L1 constructs in mouse ESCs were conducted by plating $4 \times 10^5$ cells per 35 mm diameter well onto gelatin-coated tissue culture plates and transfecting 18 hr later with Lipofectamine 2000 (Invitrogen) using 1 µg plasmid DNA per well and OptiMEM (Invitrogen) according to the manufacturer instructions. Media was replaced after 8 hr and transfected mouse ESCs passaged into a gelatin-coated 100 mm tissue culture plate 24 hr later. 200 µg/ml G418 selection for 12 days was initiated after an additional 24 hr, and drug-resistant foci fixed, stained and counted as described for HeLa cells. Independent transfections were used as biological replicates, and assays using *mneoI* or *mblastI* constructs were performed in duplicate to allow clear and consistent effects on retrotransposition rate to be detected.

Retrotransposition assays with mEGFPI tagged L1 constructs in cultured HEK293T cells were performed as described (*Goodier et al., 2013*; *Wei et al., 2000*). $2 \times 10^5$ HEK293T cells were plated in a 35 mm diameter well, then transfected with Lipofectamine 2000 (Invitrogen) and 1 µg plasmid DNA per well using OptiMEM (Invitrogen) following the manufacturer instructions 20 hr later. After a further 24 hr, fresh media was added and 48 hr later media containing 5 µg/ml puromycin (Sigma) was added daily for 7 days to select for transfected cells. Cells were collected by trypsinization and the percentage of EGFP-expressing cells determined using a FACSCanto II flow cytometer (BD Biosciences). Transfection with mutant L1 plasmid (99-gfp-JM111 or 99-gp-L1SMmut2) allowed a threshold to be established for background fluorescence. Independent transfections were used as biological replicates, and assays using mEGFPI constructs were performed in triplicate to allow clear and consistent effects on retrotransposition rate to be detected.

## Confocal microscopy

$1 \times 10^5$ U2OS cells were plated in 35 mm diameter wells, then 20 hr later transfected with Fugene6 (Promega) and 1 µg plasmid DNA per well using OptiMEM (Invitrogen) following the manufacturer instructions. Media was replaced 20 hr after transfection and cells allowed to grow for a total of 36 hr. Next, the transfected cells were trypsinized and 25–50% plated on a 15 mm diameter sterile circular polysterene coverslip in a 35 mm diameter well. 12 hr later, cells were fixed with 4% paraformaldehyde at room temperature for 30 min, permeabilized with PBS containing 0.1% (v/v) Triton X-100, then incubated with blocking solution (10% normal goat serum, 0.5% Triton-X-100 in PBS) for 30 min. After two washes in PBS containing 0.1% goat serum and 0.05% Triton X-100, coverslips were incubated with 1:1000 mouse anti-T7 primary antibody (Millipore Cat# 69522–3 RRID:AB_11211744) diluted in PBS containing 1% normal goat serum and 0.5% Triton-X-100 at 4°C overnight in a humidified chamber. Coverslips were then washed three times with PBS containing 1% normal goat serum and incubated with 1:1000 Alexa Fluor-conjugated goat anti-mouse secondary antibodies (Life Technologies) for 30 min at room temperature. Coverslips were then washed twice and mounted with SlowFade Gold antifade with DAPI (ThermoFisher) and sealed with nail polish. Slides were imaged using a Zeiss LSM-710 confocal microscope (Leica), an Axio Imager A1 Microscope (Zeiss) and captured images analyzed with ZEN lite software (Zeiss).

## Acknowledgements

We thank Sandy Martin (University of Colorado), Alex Bortvin (Carnegie Institute, Baltimore), John V. Moran (Universiy of Michigan), Jef D. Boeke (New York University), John Goodier (Johns Hopkins University, Baltimore), Norihiro Okada (National Cheng Kung University), Niki Gray, Matt Brook, Sara Heras, and Mark Ditzel (all University of Edinburgh) for generously sharing reagents and technical expertise, the flow cytometry, imaging and animal facilities for technical support, and Wendy Bickmore and Javier Caceres (both University of Edinburgh) for critical comments on the manuscript.

## Additional information

### Competing interests

OW: Reviewing editor, *eLife*. The other authors declare that no competing interests exist.

### Funding

| Funder | Grant reference number | Author |
|---|---|---|
| Medical Research Council | MC_PC_U127580973 | Marie MacLennan<br>Judith Reichmann<br>Christopher J Playfoot<br>Abigail R Mann<br>Karen Dobie<br>David Read<br>Chao-Chun Hung<br>Ian R Adams |
| Howard Hughes Medical Institute | International Career Scientist IECS-55007420 | Jose Luis García-Pérez |
| Wellcome | ISSF3 (University of Edinburgh) | Jose Luis García-Pérez |
| Max-Planck-Gesellschaft | | Elena Khazina<br>Gabriele Wagner<br>Oliver Weichenrieder |
| Kreftforeningen | 2293664-2011 | Ragnhild Eskeland |
| Universitetet i Oslo | | Ragnhild Eskeland |
| Ministerio de Economía y Competitividad | FIS-FEDER-PI11/01489 and FIS-FEDER-PI14/02152 | Jose Luis García-Pérez |
| H2020 European Research Council | ERC-Consolidator ERC-STG-2012-233764 | Jose Luis García-Pérez |
| Seventh Framework Programme | ERA Network Neuron II PCIN-2014-115-ERA-NET | Jose Luis García-Pérez |
| Medical Research Council | MC_PC_U127574433 | Richard R Meehan |
| Medical Research Council | Doctoral Training Awards | Judith Reichmann<br>Christopher Playfoot |
| Consejería de Economía, Innovación, Ciencia y Empleo, Junta de Andalucía | CICE-FEDER-P12-CTS-2256 | Marta García-Cañadas<br>Carmen Salvador-Palomeque<br>Paula Peressini<br>Laura Sanchez<br>Jose Luis García-Pérez |

The funders had no role in study design, data collection and interpretation, or the decision to submit the work for publication.

### Author contributions

MM, Formal analysis, Investigation, Methodology, Writing—review and editing, Investigation: Ubiquitylation assays, TUBE2 pull-downs, isolation of *TEX19.1*-YFP complexes, co-immunoprecipitations, phenotypic analysis of *Ubr2*$^{-/-}$ mice, phenotypic analysis of *Tex19.1*$^{-/-}$ ES cells, translation luciferase assays; MG-C, Data curation, Formal analysis, Investigation, Methodology, Writing—review and editing, Investigation: Retrotransposition assays in ESCs, confocal microscopy, analysis of hESCs and EC cells, phenotypic analysis of *Tex19.1*$^{-/-}$ ESCs, promoter luciferase assays; JR, Formal analysis,

Investigation, Writing—review and editing, Investigation: Phenotypic analysis of *Tex19.1*$^{-/-}$ mice, polysome gradients, oligod(T) pull-downs, size exclusion chromatography; EK, Investigation, Writing—review and editing, Investigation: Strep pull-downs from bacterial lysates; GW, Investigation, Investigation: Strep pull-downs from bacterial lysates; CJP, Investigation, Writing—review and editing, Investigation: qRT-PCR analyses of *Tex19.1*$^{-/-}$ and *Ubr2*$^{-/-}$ mice; CS-P, Investigation, Investigation: Retrotransposition assays in HeLa and U2OS cells; ARM, Resources, Investigation: Generation of *Tex19.1*$^{-/-}$ ESCs; PP, Investigation, Investigation: Retrotransposition assays in HEK293T cells; LS, Investigation, Investigation: Human and mouse ESC culture and characterization, nucleic acid extraction; KD, Investigation, Investigation: Phenotypic analysis of *Tex19.1*$^{-/-}$ ESCs; DR, Resources, Resources: Mouse colony management and tissue dissection; C-CH, Resources, Resources: Generation of *TEX19.1*-YFP ESC lines; RE, Investigation, Writing—review and editing, Investigation: Size exclusion chromatography; RRM, Conceptualization, Writing—review and editing; OW, Conceptualization, Supervision, Writing—review and editing; JLG-P, IRA, Conceptualization, Supervision, Writing—original draft, Writing—review and editing

### Author ORCIDs
Marie MacLennan, http://orcid.org/0000-0001-8265-7003
Ragnhild Eskeland, http://orcid.org/0000-0003-2789-3171
Oliver Weichenrieder, http://orcid.org/0000-0001-5818-6248
Jose Luis García-Pérez, http://orcid.org/0000-0002-8132-9849
Ian R Adams, http://orcid.org/0000-0001-8838-1271

### Ethics
Animal experimentation: Mouse experiments were performed in accordance with local ethical guidelines and under authority of UK Home Office Project Licence PPL 60/4424.

---

## Additional files

### Supplementary files
• Supplementary file 1. Preliminary mass spectrometry data from TEX19.1-YFP immunoprecipitates. Preliminary mass spectrometry data obtained from a single immunoprecipitation and mass spectrometry experiment from ESC cytoplasmic lysates. TEX19.1-YFP and YFP alone immunoprecipitations were run on a polyacrylamide gel and each gel lane cut into seven bands according to size (T1-T7). Proteins identified by mass spectrometry in TEX19.1-YFP immunoprecipitates, but not in YFP controls, are listed in the table. This preliminary list could contain proteins that interact non-specifically with the beads or YFP that are variably detected in YFP alone samples, and/or proteins that are variably present in TEX19.1-YFP samples that cannot be consistently reproduced under these experimental conditions. Proteins that were verified as TEX19.1 interactors in independent samples by Western blotting are listed in *Supplementary file 2*.

• Supplementary file 2. Proteins identified in TEX19.1-YFP immunoprecipitates. Proteins identified by mass spectrometry in TEX19.1-YFP immunoprecipitates from mouse ESC cytoplasmic lysates, but not in YFP controls. Only interactors verified by Western blotting (*Figure 2*) are listed. Queries matched indicates the number of MS/MS spectra that were matched to each protein, coverage indicates the percentage of target protein matched by MS/MS spectra.

• Supplementary file 3. Oligonucleotides used in this study. Lower case nucleotides in the repair template sequence indicate mutations relative to wild-type sequence.

• Supplementary file 4. Plasmids used in this study. Description of plasmids used in this study.

• Supplementary file 5. Antibodies used for western blots. List of antibodies, sources and dilutions used for Western blots.

## Major datasets

The following previously published dataset was used:

| Author(s) | Year | Dataset title | Dataset URL | Database, license, and accessibility information |
|---|---|---|---|---|
| Lin S, Lin Y, Nery JR, Urich MA, Breschi A, Davis CA, Dobin A, Zaleski C, Beer MA, Chapman WC, Gingeras TR, Ecker JR, Snyder MP | 2014 | Comparison of the transcriptional landscapes between human and mouse tissues | https://www.ncbi.nlm.nih.gov/geo/query/acc.cgi?acc=GSE36025 | Publicly available at the NCBI Gene Expression Omnibus (accession no: GSE36025) |

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
