## [Decision Letter]

Thank you for submitting your article "Mobilization of LINE-1 Retrotransposons is
Restricted by *Tex19.1* in Mouse Embryonic Stem Cells" for
consideration by *eLife*. Your article has been reviewed by three peer
reviewers, and the evaluation has been overseen by Marianne Bronner as Reviewing and
Senior Editor. Reviewer 2 is Miguel Branco and reviewer 3 is Todd Macfarlan. The
reviewers have discussed the reviews with one another and the Reviewing Editor has
drafted this decision to help you prepare a revised submission. At the request of
reviewer 1, who chose not to reveal their identity, we have included a separate review,
though the consolidated review merges the salient points of all three reviewers.

Summary:

In this paper MacLennan et al. introduce new insights into the role of Tex19.1 protein,
a previously identified restriction factor for retrotransposon activity in mammals. The
authors show post-transcriptional suppression of L1 retrotransposons in multiple cell
lines, and reveal that this involves ubiquitination of the L1 ORF1p protein. A role for
a cognate UBR2 E3 ligase is identified. It is demonstrated that *Tex19.1*
mediates levels of ORF1p and retrotransposition, including in mouse ES cells. This is
the first report of modifications of ORF1p in the germline and of its control by
ubiquitination. As such, this is a significant paper and worthy of publication in
*eLife*. It is a well-dissected mechanism, and a convincing
demonstration of the role of TEX19.1 in restricting L1 retrotransposition.

Major revisions:

1) Subsection “TEX19.1 Orthologs Directly Interact With L1-ORF1p”; Discussion, second
paragraph. The authors claim direct binding of TEX19, UBR2, and ORF1p. Of the more than
100 proteins identified by various studies as associated with human L1 ORF1p, almost all
associations disappear upon RNase treatment. Interactions between bacterially-expressed
proteins in pull-down assays may also be mediated by RNA-tethering. While the ability to
interact with an RNA-binding deficient mutant of ORF1p is telling, I do not regard it as
conclusive, especially when protein interactions are weak as noted. The most obvious
control has not been reported. Therefore, IP assays should be performed in the presence
of RNases before the statement is made that *Tex19.1* is the first
example of a mammalian host protein directly binding an L1 protein.

2) In the experiments of the subsection “UBR2 Interacts With L1-ORF1p And Regulates L1
Independently Of *Tex19.1* Orthologs”, the effect of Ubr2 is uncoupled
from those of *Tex19.1*. However, the authors propose a model in which
"TEX19.1 stimulates a basal activity of UBR2 to bind to and polyubiquitylate
L1-ORF1p (Figure 7)". Therefore, it should
be directly tested if UBR2 mediated binding and polyubiquitination of ORF1p is in fact
increased in the presence of overexpressed TEX19. Similarly, it would be simple to test
whether *Tex19.1* KO cells show reduced ubiquitylation and turnover of
ORF1p. This would be strongly suggestive evidence that the proposed mechanism works
endogenously in ESCs.

3) Subsection “TEX19.1 Interacts With Multiple Components Of The Ubiquitin-Proteasome
System”, last paragraph; subsection “Isolation Of TEX19.1-YFP Complexes”. To identify
proteins interacting with Tex19.1 in mouse ES cells, IP gel lanes were excised in
sections and subjected to MS sequencing. Even if not relevant to the conclusions of this
paper, I believe the complete results of these MS analyses should be summarized in a
supplemental Table for the information of future *Tex19.1* investigations
(and especially in light of the recent Tarabay et al. paper). At this time, only those
proteins chosen for Western blot confirmation are noted in [Supplementary-material SD1-data].

4) One basic premise of the paper is that TEX19.1 acts on L1s post-translationally and
does not affect L1 transcript abundance, but the relevant data are on the weak side.
Firstly, it looks like only one litter of a Tex19.1 het x het cross was analyzed, which
is not ideal. Then there's no information on what the qPCR primers are targeting – ORF1,
ORF2 or the 5' UTR? If the latter, which class? If the former two, then it remains
possible that TEX19.1 only affects full-length elements. The best experiment for this
would be a Northern blot, but a qPCR to the 5' UTR of active L1 classes would probably
be sufficient.

5) Does endogenous UBR2 associate with TEX19 in ESCs (CoIP)?

6) Are there particular domains of *Tex19.1*/Orf1p that are critical for
their interaction? Are these domains conserved amongst human and mouse orthologs?
(showing alignments of human and mouse *Tex19.1* and orf1p would be
helpful).

7) Is Ubr2 required for *Tex19* effects on Orf1p? This could be achieved
by overexpressing *Tex19* in primary cell lines from Ubr2 mutants (like
NSCs or PMEFs).

*Reviewer 1:*

In this paper MacLennan et al. introduce new insights into the role of
*Tex19.1* protein, a previously identified restriction factor for
retrotransposon activity in mammals. The authors show post-transcriptional suppression
of L1 retrotransposons in multiple cell lines, and reveal that this involves
ubiquitination of the L1 ORF1p protein. A role for a cognate UBR2 E3 ligase is
identified. It is demonstrated that *Tex19.1* mediates levels of ORF1p
and retrotransposition, including in mouse ES cells. This is the first report of
modifications of ORF1p in the germline and of its control by ubiquitination. As such,
this is a significant paper and worthy of publication in *eLife*.

A few modest additions would improve the clarity and impact of the paper:

Subsection “TEX19.1 Orthologs Directly Interact With L1-ORF1p”; Discussion, second
paragraph. The authors claim direct binding of TEX19, UBR2, and ORF1p. Of the more than
100 proteins identified by various studies as associated with human L1 ORF1p, almost all
associations disappear upon RNase treatment. Interactions between bacterially-expressed
proteins in pull-down assays may also be mediated by RNA-tethering. While the ability to
interact with an RNA-binding deficient mutant of ORF1p is telling, I do not regard it as
conclusive, especially when protein interactions are weak as noted. The most obvious
control has not been reported. Therefore, IP assays should be performed in the presence
of RNases before the statement is made that Tex19.1 is the first example of a mammalian
host protein directly binding an L1 protein.

In the experiments of the subsection “UBR2 Interacts With L1-ORF1p And Regulates L1
Independently Of *Tex19.1* Orthologs”, the effect of Ubr2 is uncoupled
from those of Tex19.1. However, the authors propose a model in which "TEX19.1
stimulates a basal activity of UBR2 to bind to and polyubiquitylate L1-ORF1p (Figure 7)". Therefore, it should be directly
tested if UBR2 mediated binding and polyubiquitination of ORF1p is in fact increased in
the presence of overexpressed TEX19.

Subsection “TEX19.1 Interacts With Multiple Components Of The Ubiquitin-Proteasome
System”, last paragraph; subsection “Isolation Of TEX19.1-YFP Complexes”. To identify
proteins interacting with Tex19.1 in mouse ES cells, IP gel lanes were excised in
sections and subjected to MS sequencing. Even if not relevant to the conclusions of this
paper, I believe the complete results of these MS analyses should be summarized in a
supplemental Table for the information of future Tex19.1 investigations (and especially
in light of the recent Tarabay et al. paper). At this time, only those proteins chosen
for Western blot confirmation are noted in [Supplementary-material SD1-data].

Subsection “*Tex19.1* Orthologs Stimulate Polyubiquitination and
Degradation of L1-ORF1p”. Were any attempts made to map the ubiquitinated residues in
ORF1p, perhaps by MS analyses? While I do not regard such data as critical to the
submission, any attempts made could be reported. Belancio et al. (NAR 2014 v. 42)
previously mentioned 32 predicted ubiquitinated sites in the C-terminus of ORF1p.
Reporting this fact would be of interest (and suggest how difficult mapping of
ubiquitinated sites might become for ORF1p).

---

## [Author Response]

*Major revisions:*

*1) Subsection “TEX19.1 Orthologs Directly Interact With L1-ORF1p”; Discussion,
second paragraph. The authors claim direct binding of TEX19, UBR2, and ORF1p. Of the
more than 100 proteins identified by various studies as associated with human L1
ORF1p, almost all associations disappear upon RNase treatment. Interactions between
bacterially-expressed proteins in pull-down assays may also be mediated by
RNA-tethering. While the ability to interact with an RNA-binding deficient mutant of
ORF1p is telling, I do not regard it as conclusive, especially when protein
interactions are weak as noted. The most obvious control has not been reported.
Therefore, IP assays should be performed in the presence of RNases before the
statement is made that Tex19.1 is the first example of a mammalian host protein
directly binding an L1 protein.*

We agree with the reviewers that additional data supporting this important point would
improve the study as most reported L1-ORF1p interactions are mediated by RNA. However,
these interaction partners usually failed to bind L1-ORF1p when coexpressed in bacteria.
To be sure about the direct interaction between TEX19 and L1-ORF1p, we have repeated the
pulldown experiments with the bacterially expressed human proteins in the presence of
micrococcal nuclease, which digests both RNA and DNA, and we observe no difference in
the efficiency of the pulldown (Figure 3). As
DNaseI was present in the bacterial lysis buffer as well, we conclude that the observed
interaction is independent of nucleic acids.

We have moved panel G from Figure 2 (interactions
between bacterially-expressed proteins) and included this as part of an additional
figure (new Figure 3) along with the micrococcal
nuclease data (Figure 3) and new data to address
major point 6 (see below). We have also updated the Results and Methods text to reflect
this.

*2) In the experiments of the subsection “UBR2 Interacts With L1-ORF1p And
Regulates L1 Independently Of Tex19.1 Orthologs”, the effect of Ubr2 is uncoupled
from those of Tex19.1. However, the authors propose a model in which "TEX19.1
stimulates a basal activity of UBR2 to bind to and polyubiquitylate L1-ORF1p (Figure 7)". Therefore, it should be directly
tested if UBR2 mediated binding and polyubiquitination of ORF1p is in fact increased
in the presence of overexpressed TEX19. Similarly, it would be simple to test whether
Tex19.1 KO cells show reduced ubiquitylation and turnover of ORF1p. This would be
strongly suggestive evidence that the proposed mechanism works endogenously in
ESCs.*

We appreciate the reviewer’s point that the text is a bit loose here and we cannot
distinguish increased binding, increased polyubiquitination, or other potential
mechanisms that increase the activity of UBR2. We have tightened this up to read
“TEX19.1 stimulates a basal activity of UBR2 that promotes polyubiquitination of
L1-ORF1p (Figure 7)”. We have also tried to make
this clear in the legend for the model shown in Figure 7 by adding the text – “We cannot exclude the possibility that additional
factors and/or mechanisms contribute to the effects of UBR2 and TEX19.1 on the stability
of mL1-ORF1p.”

We have not tested whether UBR2 binding to L1-ORF1p increases in response to
overexpressed TEX19, but this experiment probably requires purification of pure
populations of UBR2:L1-ORF1p and UBR2:TEX19:L1-ORF1p complexes to accurately determine
binding affinities in vitro. As expression of TEX19 stimulates polyubiquitination and
turnover of L1-ORF1p in cells this will confound any analysis of UBR2-L1-ORF1p
interactions by quantitative co-IP. We did however show increased polyubiquitination of
L1-ORF1p in response to overexpressed TEX19 in Figure 3 of the original submission (now Figure 4).

Similarly, quantitatively precipitating transient polyubiquitylated species of the
heterogenous copies of L1-ORF1p endogenously expressed in ES cells is a difficult
experiment. Our preliminary data from control ES cells (original Figure 6, now Figure 7)
indicated some variability in the extent of the heterogenous polyubiquitin smear which
may reflect endogenous deubiquitylase activity in the ES cell lysates. We note that the
cell-based ubiquitylation assay used in Figure 3
in the original submission (now Figure 4) is
performed under denaturing conditions to inactivate deubiquitylase enzymes.

With respect to turnover of ORF1p in *Tex19.1^-/-^* mouse ES
cells, we showed in the original manuscript that *Tex19.1* KO ES cells
have increased abundance of L1-ORF1p protein, with no detectable change in L1 RNA
abundance, or activity of reporters for either L1 promoters or L1 translation sequences
(original submission Figure 6 and Figure 6—figure
supplement 3, now Figure 7 and Figure 7—figure supplement 3). Taken together
these data suggest that a change in L1-ORF1p turnover is contributing to the increased
L1-ORF1p abundance in *Tex19.1* KO ES cells. However, we cannot exclude
the possibility that the change in L1-ORF1p abundance reflects the behaviour of variant
L1 sequences that were not captured in the qRT-PCR or reporter assays. We have therefore
included an additional paragraph in the Discussion discussing stating this
possibility.

*3) Subsection “TEX19.1 Interacts With Multiple Components Of The
Ubiquitin-Proteasome System”, last paragraph; subsection “Isolation Of TEX19.1-YFP
Complexes”. To identify proteins interacting with Tex19.1 in mouse ES cells, IP gel
lanes were excised in sections and subjected to MS sequencing. Even if not relevant
to the conclusions of this paper, I believe the complete results of these MS analyses
should be summarized in a supplemental Table for the information of future Tex19.1
investigations (and especially in light of the recent Tarabay et al. paper). At this
time, only those proteins chosen for Western blot confirmation are noted in [Supplementary-material SD1-data].*

We have included a file showing the TEX19.1-YFP specific MS hits for the reviewers.

*4) One basic premise of the paper is that TEX19.1 acts on L1s
post-translationally and does not affect L1 transcript abundance, but the relevant
data are on the weak side. Firstly, it looks like only one litter of a Tex19.1 het x
het cross was analyzed, which is not ideal. Then there's no information on what the
qPCR primers are targeting – ORF1, ORF2 or the 5' UTR? If the latter, which class? If
the former two, then it remains possible that TEX19.1 only affects full-length
elements. The best experiment for this would be a Northern blot, but a qPCR to the 5'
UTR of active L1 classes would probably be sufficient.*

Data on L1 transcript abundance in *Tex19.1^-/-^* testes have
been reported in multiple previous publications using different assays and primer sets:
L1 transcript abundance does not change in *Tex19.1^-/-^* testes
(Öllinger et al., 2008, Reichmann et al., 2012, Tarabay et al., 2013). There has clearly
been more than a single litter of animals analysed in terms of L1 RNA abundance across
all these studies, which encompasses multiple genetic backgrounds. We have also assessed
abundance of H3K4me3, a chromatin modification associated with transcription, at L1
sequences in *Tex19.1^-/-^* testes and this does not change
either (Crichton et al., 2017, bioRxiv, 099119).

For L1-ORF1p protein abundance, our apologies, we could have made this clearer. L1-ORF1p
abundance has not only been examined in one litter. These data are representative of
seven *Tex19.1^-/-^* animals across four litters. We have
included this information in the legend to Figure 1. The confirmatory immunofluorescence in Figure 2 represents an additional three
*Tex19.1^-/-^* animals from two litters that were not
analysed by Western blot. The use of littermate controls for these experiments is
particularly important when analysing RNAs or proteins that exhibit developmentally
dynamic expression patterns: there is less variation in developmental stage within
litters than between litters, which means fewer animals are required to detect a
statistically significant effect (n=3 in Figure 2).

We agree with the reviewers that it is important to determine whether full-length L1
RNAs are affected by *Tex19.1* The primers used for L1 qRT-PCR in the
original submission were directed against ORF2 as this set of primers is normally used
to detect most L1 RNAs present in cells. This was stated in the primer list in [Supplementary-material SD2-data] of the
original submission. But we agree this could have been made clear in the figure legends
also. In response to the reviewer’s comments, we have included this information in the
legends to Figure 1 and 7. In addition, we have
also included new qRT-PCR data on testes for the RNA abundance A, T_f_ and
G_f_ L1 subtypes using primers designed against the 5’UTR of active subtypes
of L1. These data indicate that these active L1 subtypes are expressed at similar levels
in *Tex19.1^-/-^* and control testes.

Finally, please note that it is possible that the heterogeneous nature of L1 DNA, RNA
and protein populations means that changes in L1-ORF1p abundance could be caused by
changes in the abundance of variant L1 sequences that were not detected in the qRT-PCR
assays (or would not be detected by Northern blot). Therefore marked/tagged copies of L1
are needed to demonstrate that *TEX19* can affect L1-ORF1p abundance
without altering the abundance of its encoding L1 RNA molecule. These data are in Figure 3 (now Figure 4). We have tried to stress the importance of these data more in the
Discussion.

*5) Does endogenous UBR2 associate with TEX19 in ESCs (CoIP)?*

It is not clear whether the reviewer is referring to mouse or human proteins. If mouse,
the UBR2 co-immunoprecipitating with TEX19.1-YFP in ESCs in Figure 2 and Figure 2 is
endogenous UBR2. The interaction between endogenous UBR2 and endogenous TEX19.1 has been
shown previously in mouse testes (Yang et al., 2010). If human, our reagents are
probably not suitable to test this, the commercial anti human TEX19 antibodies appear to
strongly cross-react with other proteins (see Figure 9).

Author response image 1.Western blot of endogenous TEX19 in *UBR2* mutant and
control HCT116 cells.Parental HCT116 cells and three independent clones isolated from either
*UBR2* or control CRISPR genome editing experiments are
shown. UBR2 CRISPR guides were directed against exon 2 of
*UBR2*, and the three independent *UBR2* mutant
clones all have deletions in the UBR domain (top panel). Endogenous TEX19
protein is readily detected in parental and control CRISPR lines, but not in
*UBR2* mutant lines (bottom panel. A cross-reacting band
detected with the TEX19 antibodies is marked with an asterisk, and antibodies
to lamin B were used as a loading control.**DOI:**
http://dx.doi.org/10.7554/eLife.26152.027

*6) Are there particular domains of Tex19.1/Orf1p that are critical for their
interaction? Are these domains conserved amongst human and mouse orthologs? (showing
alignments of human and mouse Tex19.1 and orf1p would be helpful).*

We acknowledge the reviewer’s suggestion and agree that this is useful and informative.
Alignments for L1-ORF1p and Tex19 have been published by Boissinot et al. (2016) and
Bianchetti et al. (2015) and these are summarized in a new figure (Figure 3, panels C and D). Furthermore, and as suggested by the
reviewers, we have performed additional pulldown experiments with sub-fragments of the
two proteins to further confine their direct interaction (new figure, Figure 3). These experiments reveal that the
highly conserved N-terminal MCP region of Tex19 is necessary and sufficient to interact
with L1-ORF1p and suggest the conserved, C-terminal half of the L1ORF1p coiled coil
domain as a likely interaction site in all mammals.

*7) Is Ubr2 required for Tex19 effects on Orf1p? This could be achieved by
overexpressing Tex19 in primary cell lines from Ubr2 mutants (like NSCs or
PMEFs).*

Published data in the field shows that TEX19.1 protein is unstable and not detectable in
the absence of UBR2 in mouse testes (Yang et al., 2010). We performed a similar
experiment to the one suggested by the reviewers but took advantage of endogenous TEX19
expression in HCT116 cells rather than using an overexpression experiment. These
preliminary experiments showed that TEX19 protein becomes unstable and undetectable in
the UBR2 mutant lines (Figure 9).
Therefore, the requirement for UBR2 to maintain TEX19.1 protein stability, and therefore
presumably *Tex19.1* function (Yang et al., 2010), in mouse germ cells
extends to human somatic cells. However, as might be expected, endogenous expression of
L1-ORF1p is difficult to detect in somatic cell lines (Philippe et al., 2016,
*ELife*, 13926), and the slow growth and altered cell cycle profiles
of the UBR2 mutant lines, which presumably reflects the known functions of this protein
in maintaining genome integrity (Ouyang et al., 2006), makes meaningful comparisons with
control cells difficult.